# A new hypothesis for the origin of Amazonian Dark Earths

Lucas C. R. Silva [1,2✉], Rodrigo Studart Corrêa[3], Jamie L. Wright[1], Barbara Bomfim [1,4], Lauren Hendricks [2], Daniel G. Gavin[2], Aleksander Westphal Muniz [5], Gilvan Coimbra Martins[5], Antônio Carlos Vargas Motta[6], Julierme Zimmer Barbosa [7], Vander de Freitas Melo[6], Scott D. Young[8], Martin R. Broadley [8] & Roberto Ventura Santos[9]

Amazonian Dark Earths (ADEs) are unusually fertile soils characterised by elevated concentrations of microscopic charcoal particles, which confer their distinctive colouration. Frequent occurrences of pre-Columbian artefacts at ADE sites led to their ubiquitous classification as Anthrosols (soils of anthropic origin). However, it remains unclear how indigenous peoples created areas of high fertility in one of the most nutrient-impoverished environments on Earth. Here, we report new data from a well-studied ADE site in the Brazilian Amazon, which compel us to reconsider its anthropic origin. The amounts of phosphorus and calcium—two of the least abundant macronutrients in the region—are orders of magnitude higher in ADE profiles than in the surrounding soil. The elevated levels of phosphorus and calcium, which are often interpreted as evidence of human activity at other sites, correlate spatially with trace elements that indicate exogenous mineral sources rather than in situ deposition. Stable isotope ratios of neodymium, strontium, and radiocarbon activity of microcharcoal particles also indicate exogenous inputs from alluvial deposition of carbon and mineral elements to ADE profiles, beginning several thousands of years before the earliest evidence of soil management for plant cultivation in the region. Our data suggest that indigenous peoples harnessed natural processes of landscape formation, which led to the unique properties of ADEs, but were not responsible for their genesis. If corroborated elsewhere, this hypothesis would transform our understanding of human influence in Amazonia, opening new frontiers for the sustainable use of tropical landscapes going forward.

[1] Environmental Studies Program, University of Oregon, Eugene, OR, USA. [2] Department of Geography, University of Oregon, Eugene, OR, USA. [3] Environmental Sciences Program - PPGCA/FUP, University of Brasília, Planaltina, DF, Brazil. [4] Lawrence Berkeley National Laboratory, Berkeley, CA, USA. [5] Brazilian Agricultural Research Corporation - CPAA/Embrapa Amazônia Ocidental, Manaus, AM, Brazil. [6] Department of Soil Science, University of Paraná, Curitiba, PR, Brazil. [7] Federal Institute of Southeast Minas Gerais, Barbacena, Minas Gerais, Brazil. [8] School of Biosciences, University of Nottingham, Nottingham, UK. [9] Institute of Geosciences, University of Brasília, Brasília, DF, Brazil. ✉email: lsilva7@uoregon.edu

Discovered decades ago in central Brazil[1], Amazonian Dark Earths (ADEs) are now regarded as a Pan-Amazonian anthropological phenomenon[2]. The exceptional fertility of ADEs, combined with a high concentration of pyrogenic carbon[3], and frequent observations of pre-Columbian artefacts led to their ubiquitous classification as anthropic soils[4], often referred to as Pretic Anthrosols[5]. The current paradigm of ADE formation postulates that pyrolysis of nutrient-rich biomass created hundreds of high-fertility patches (some as large as 350 hectares)[4] surrounded by the acidic dystrophic soils[6] that covers most of the Amazon basin[7]. However, neither the timing nor the magnitude of human occupation needed for ADE formation is well-defined. Among key questions that remain unanswered is how indigenous populations—subsisting from incipient agriculture, aquatic wildlife, and hunting—created persistent areas of high fertility in the notoriously nutrient-impoverished Amazon basin?

Amazonian landscapes are dominated by highly weathered Oxisols and Ultisols, which are characterised by high acidity and low nutrient concentrations. As in other parts of the tropics, native plant species display adaptations, such as nutrient uptake from litter and tolerance to elements that are toxic for crop species[8–10], which allow Amazonian soils to maintain some of the most productive ecosystems on Earth while limiting food production even under intensive management[11]. Recent findings suggest that domestication of native plant species dates back to >10,000 years ago in western Amazonia[12], with the emergence of complex societies that relied on soil management for agriculture occurring <4000 years ago[13]. Some of the earliest evidence of sedentary settlements in Amazonia comes from the Peruvian Andes, where records of deforestation and soil erosion exist for much of the past 6900 years[14]. For most of the Amazon basin, however, the earliest evidence of intensive cultivation (inferred from pollen data) falls between 3380 and 700 years ago[15]. Notably, ADEs are rare near Andean settlements[16] and the vast majority of ADE sites are found thousands of kilometres away in the central and eastern Brazilian Amazon[17], where evidence of management is even more recent (2500–500 years ago).

The existence of small and scattered ADE patches (<1–2 ha) in central and eastern Amazonia is often attributed to transient communities containing few people over a long time. By contrast, larger ADE patches are thought to have been created by people living in large sedentary villages[18]. Informed by anthropological studies, we sought to develop new evidence from well-studied ADE sites, which is thought to have originated from a large settlement <2000 years ago (Fig. 1). The site is home to the Brazilian Agroforestry Research Station (EMBRAPA – CPAA), where the local ADE has been classified as a typical Anthrosol. As in other ADEs, archaeological artefacts are found in charcoal-rich soil layers in which pyrogenic carbon is 5–10% of the total carbon pool, and where vestiges of human faeces are also present[19]. It is clear that pre-Columbian land-use occurred in our region, but it remains unclear how ancient (or modern) soil management can create such areas of high fertility. The current view of ADE genesis suggests that the artefacts found in nutrient- and charcoal-rich Pretic horizons result from biomass burning and application to the soil. This view has fuelled an entire industry of charcoal production from biomass (i.e. biochar) in which ADEs are described as a model for sustainable agriculture[20]. Under experimental application, however, biochar alone (or in combination with fertilisers) has proved inadequate to replicate basic characteristics of indigenous ADEs, such as their long-lasting mineral fertility[21–23]. This reveals a lack of understanding that warrants further investigation into the genesis of ADEs.

To improve understanding of ADE formation, we used EMBRAPA's reference profiles across a contiguous patch of ~12 ha to select ~4.5 ha for intensive sampling in a paired scheme that

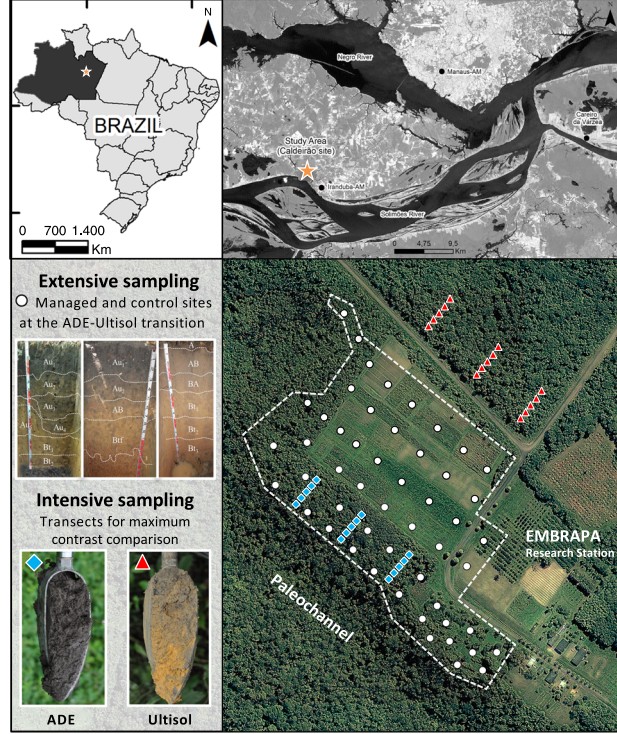

**Fig. 1 Experimental station in Iranduba, Amazonas, Brazil.** Photos depict typical soil profiles previously described as sandy clay loam ADEs and very clayey Ultisols (left to right: ADE Orthodystric; ADE Lixic, Ortheoutric; and Ultisol Hyperdystric), images adapted from ref. [5]. Auger photos depict the difference in colouration caused by differences in pyrogenic carbon content measured here. White circles indicate reference soil profiles used to generate a site-wide spatial analysis of extractable phosphorus (P) and calcium (Ca) using data from EMBRAPA-CPAA archives, shown in Fig. 2. Interpolation maps of extractable P and Ca were used to select replicated transects for maximum contrast comparison of total stocks of P and Ca in ADE (blue diamonds) and Ultisol (red triangles) profiles, shown in Fig. 3, and other mineral elements shown in Supplementary Information.

provides maximum contrast among ADE and adjacent Ultisols. All intensively sampled soils were under dense secondary evergreen rainforest (>12 m canopy height), which had not been managed for at least 40 years since the research station was established. Across the study area, we measured mineral nutrient excess in ADE to estimate the input necessary to explain its origin. As a proxy for the overall amount of mineral nutrient input, we focused on phosphorus (P) and calcium (Ca) excess, which are two of the most limiting elements in tropical landscapes[24–26]. We then quantified the association of P and Ca excess with the concentration of trace elements, including strontium (Sr) and neodymium (Nd), which serve as proxies for mineral source. Previous research showed a direct relationship between the amount of charcoal and the colour of ADEs at our site, where microscopic charcoal (<500 μm) comprises most of the pyrogenic carbon pool, and pretic horizons are dominated by very small particles (<40 μm)[5]. Building on those findings, we determined the source and timing of microcharcoal deposition using stable carbon isotope ratios and calibrated radiocarbon ($^{14}$C) dates of ADE and Ultisol samples. Finally, we analysed carbon and nutrient data in light of the local anthropological context to estimate the chronology of management and the population density needed to attain the observed gain in ADE fertility. Our results show that large sedentary populations would have had to manage soils thousands of years prior to the

emergence of agriculture in the region or, more likely, that indigenous peoples used their knowledge to identify and preferentially settle areas of exceptionally high fertility before the onset of intensive land use in central Amazonia.

## Results and discussion

**Nutrient budget.** A spatial interpolation analysis using nutrient concentrations from reference soil profiles (Fig. 1) was used to determine the sampling effort for the determination of total nutrients and stocks. A map of extractable concentration was used to guide the determination of total P and Ca stocks. Extractable P and Ca concentrations were 1–3 orders of magnitude greater within the ADE patch than in the surrounding Ultisol (Fig. 2). The well-defined spatial pattern of extractable P and Ca from the most fertile ADE profiles (blue) to nutrient-depleted Ultisols (red) guided an in-depth characterisation of elemental and isotopic composition performed along six transects ~100 m apart from one another, each containing 5 soil profiles, ~10 m apart from one another, for a total of 30 intensively sampled profiles (blue diamonds). Those profiles yielded samples for a maximum contrast comparison—that is, highest and lowest soil fertility along the site's southwest–northeast axis—on the basis of total nutrient stocks between ADEs and reference Ultisols (red triangles, Fig. 3).

To estimate total nutrient stocks, bulk density was multiplied by Ca and P concentrations at 10 cm depth increment up to 1 m in each profile. Without exception, all sampled profiles showed much larger nutrient stocks (expressed as megagrams per hectare; Mg ha$^{-1}$) in the ADE than in the surrounding soil. On average, we found that ADE profiles contain 16.8 Mg P ha$^{-1}$ and 14.9 Mg Ca ha$^{-1}$ in excess relative to the surrounding Ultisol (Fig. 3). Outside the ADE patch, the stocks of P and Ca are as low as those typically reported for other Brazilian Oxisols and Ultisols[27]. Inside the patch, the stocks of P and Ca are similar to those found in other ADEs[28] and higher than typical values for agricultural and timber production systems under systematic nutrient addition[29]. Notably, neither the elemental stoichiometries of freshwater fish (Ca:P ~2.13)[30] nor human faeces (Ca:P ~2)[31] are apparent in ADE profiles and differences in fertility go beyond those two macronutrients. Concentrations of 16 mineral elements were significantly greater in ADE than in Ultisol profiles in paired $t$-tests: Ba, Ca, Cd, Co, Cs, Cu, K, Li, Mg, Mn, Ni, P, Rb, Sr, Tl, and Zn ($p < 0.05$; Supplementary Table 1). Most chemical parameters, including trace elements, were positively correlated with P and/or Ca with a strong clustering within the Ultisols but much variability within ADEs (Principal Component Axis 1 explained ~56% of the variance in elemental composition clustered by soil type; Supplementary Fig. 1). On the other hand, a weak clustering by depth was observed with a similar range of variation for ADEs and Ultisols (Principal Component Axis 2).

**Exogenous inputs.** We found a small but significant difference in total carbon concentration (<5 g kg$^{-1}$) between ADE and Ultisol profiles near the soil surface (0–30 cm depth), but that difference disappeared at greater depths (80–100 cm; Fig. 4). Nutrient-rich dark-coloured profiles have much higher pyrogenic carbon concentrations 5- to 10-fold higher than those found in Ultisols. Stable carbon isotope ratios indicate typical forest biomass input to Ultisols, whereas C$_4$ grasses contributed 5–19% of total carbon to ADE horizons (30–100 cm). The isotopic enrichment of ADE carbon profiles coincides with microcharcoal inputs, beginning ~7630 ± 80 years ago (Fig. 5), with a clear offset between forest organic matter (Ultisols) and enriched δ$^{13}$C ratios (ADEs). The earliest input of microcharcoal to ADEs pre-dates the earliest

evidence of land management in the region and falls within a period of expansion for C$_4$-dominated grasslands in the Solimões Basin (~20,000–7500 years ago)[32]. Importantly, our $^{14}$C dates represent the most recent of a range of possible ages because inputs from human activity, as well as natural processes, may have contributed additional charcoal into the soil matrix. Although more recalcitrant and significantly older than bulk organic matter, charcoal is degradable and can be transformed from macro to micro particles under continued disturbance[23]. This could explain the accumulation of microcharcoal at intermediate depths and implies that the earliest input events identified here are, in fact, significantly older. We posit that the timing of the formation of other ADEs would be pushed back significantly following a detailed analysis of microcharcoal deposition.

Human activity could have influenced the δ$^{13}$C values both indirectly (through the removal of trees) and directly (through the introduction of C$_4$ crops such as maize). However, previous studies yielded no evidence of grass or Maize phytoliths at our site[5]. The nearest area where fossil Maize pollen is found (~600 linear kilometres to the East of our site) sets the earliest evidence of cultivation ~4000 years ago[15], which is consistent with the emergence of cultivation in other areas of the Amazon basin[33–35]. Therefore, we can rule out agriculture as the source of C$_4$-derived carbon input at the onset of ADE genesis. In some areas of the western Amazon, charcoal and pollen records suggest human use of fire ~6500 years ago[36], but in ADE sites of central and eastern Amazon, land management is thought to be much more recent. On the other hand, a synthesis of charcoal and pollen records suggest a shift from well-drained to flooded conditions (signalling the formation of lakes) before the apex pre-Columbian activity in our region[37]. At our site, markedly different elemental signatures of ADE and Ultisols indicate different flood regimes in the past with elevated concentrations of mineral elements observed near a geomorphic feature that we interpret as a natural levee. The trace elements (e.g. Ni, Rb, Ti, Zn) found at the same depths in which we see enriched P, Ca, and charcoal levels match what might be expected from alluvial deposition and suggest major exogenous inputs to ADE profiles but not to the surrounding Ultisol (Supplementary Table 1). This implies that two critical ingredients needed for ADE formation (microcharcoal, which confers it's colour and cation-exchange capacity; and mineral elements, which are in extremely low supply in the surrounding soil) were deposited before soil management and plant cultivation took place.

The timing and magnitude of charcoal and nutrient accumulation at our site match those found in sedimentary deposits that can be traced to open vegetation fires upstream, on the basis of $^{14}$C dates and δ$^{13}$C signatures[38]. Paleoflood archives (soil biogenic silica)[39] and records of monsoon intensity (speleothem oxygen and strontium isotope ratios)[40, 41] indicate a climate-driven shift in river dynamics following a persistent dry period (~8000–4000 years ago)[42], which is thought to have reduced fire disturbance, causing a regional increase in tree cover, well into the late-Holocene[43]. The resulting changes in vegetation cover could have caused divergent patterns of carbon and nutrient accumulation in flooded versus non-flooded areas[44], consistent with ADE and Ultisol differences at our site. Many areas of central Amazonia today are associated with sediment deposits representative of flood regimes that were either deactivated during the Holocene or are presently in the process of deactivation when sedimentary deposits become suitable habitats for grasslands within the rainforest[45]. The abundance of C$_4$-derived microcharcoal (19% of the total organic carbon pool) in deep ADE layers and the absence of in situ records of C$_4$ species—combined

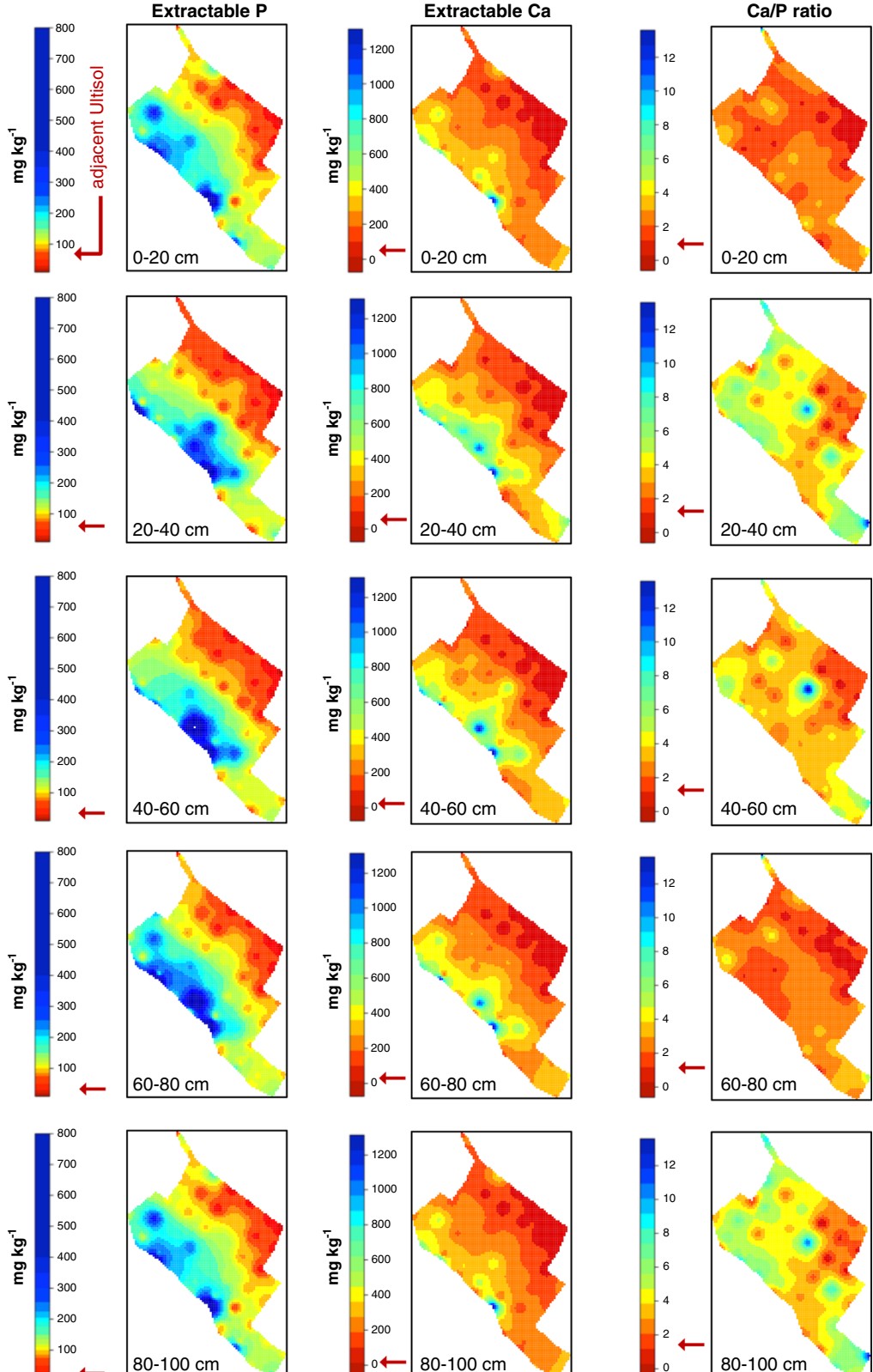

**Fig. 2 Concentration of extractable phosphorus (P) and calcium (Ca) and their ratios in ADE and Ultisol profiles.** Spatial interpolation maps developed using first-order kriging of reference profiles (white circles; Fig. 1) indicate a depositional gradient from ADEs near the paleochannel levee toward nutrient-poor Ultisols. Red arrows indicate approximate values measured in the adjacent maximum contrast Ultisol profiles under rainforest (red triangles; Fig. 1).

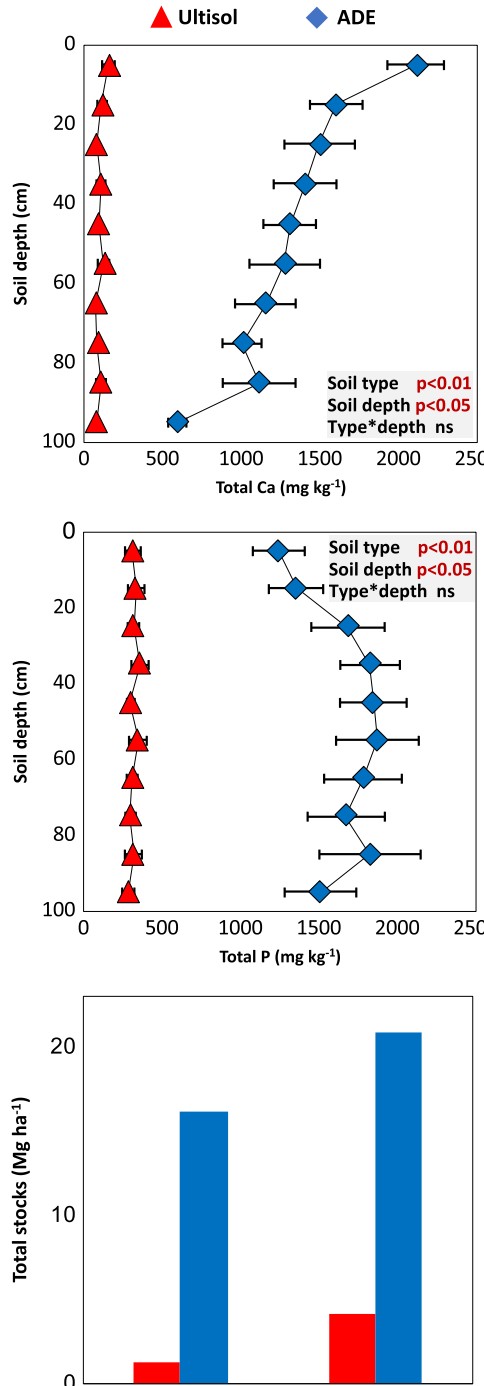

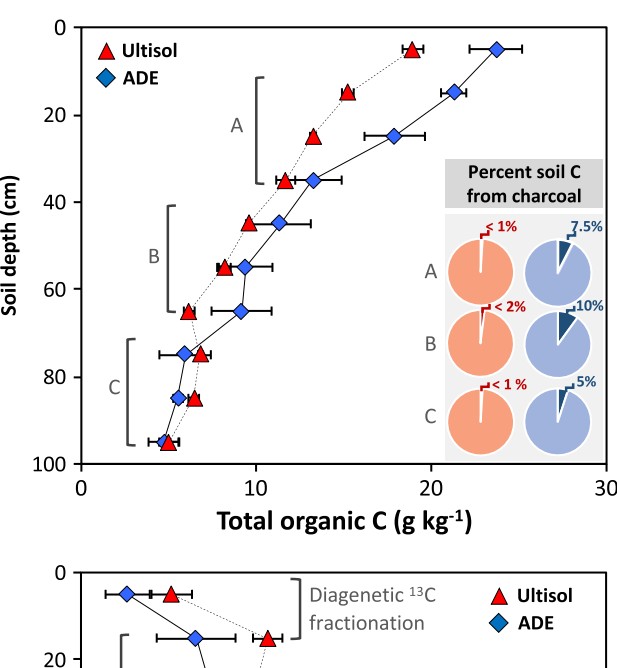

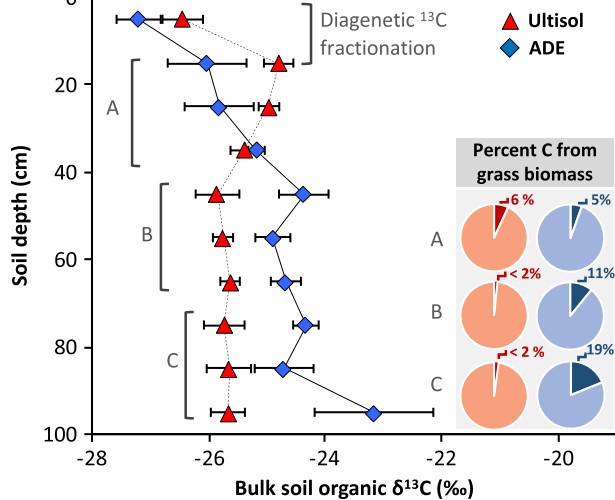

**Fig. 4 Concentration of total organic carbon and stable isotope ratios of carbon in bulk organic matter.** Stable carbon isotope ratios show an expected enrichment caused by diagenetic fractionation at the topsoil layer for both soil types, followed by a marked shift in isotopic ratios that indicates a significant contribution of grass biomass to carbon pools at deeper layers of ADE profiles, but not for the surrounding Ultisol. Data points show average values and error bars indicate two standard errors of the mean across maximum contrast profiles ($n = 5$ for ADE and $n = 5$ for Ultisol for each 10cm depth; Fig. 1). Inset panels depict the average per cent fraction of pyrogenic charcoal and grass-derived biomass in the total carbon pool (Eqs. 1 and 2) between 10 to 40cm (A), 40 to 70cm (B), and 70 to 100cm (C) depths. Data are shown in Supplementary Fig. 2.

**Fig. 3 Concentration of total phosphorus (P) and calcium (Ca) in ADE and Ultisol profiles.** Data points show average values and error bars indicate two standard errors of the mean across maximum contrast profiles ($n = 5$ for ADE and $n = 5$ for Ultisol for 10cm depth; Fig. 1). In all cases, a two-way analysis of variance shows significant differences between soil type and soil depth. The bar graph shows the total estimated stock of P and Ca up to 100cm depth calculated as the sum of average elemental concentration multiplied by bulk density at each incremental depth.

with the fact that fire frequency, charcoal production, and nutrient losses are much higher in open savannas than in rainforests[27, 29, 46]—suggest that pyrogenic carbon and nutrients were transported to our site from $C_4$-dominated landscapes upstream. Indeed, the majority of soil properties observed here could be explained by mid- to late-Holocene flood events[39],

possibly triggered by increased monsoon intensity[41], which are thought to have affected fire regime and vegetation cover throughout the region[43]. In all likelihood, such natural processes would have also contributed to the formation of other ADEs throughout the region, which could explain why a highly predictable pattern of ADE distribution emerges from maximum entropy models based solely on geomorphological parameters, such as distance from rivers and elevation[17].

Isotope ratios of the mineral soil fraction also indicate fluvial inputs to the ADE soil. We found that ADE horizons are less radiogenic ($p < 0.05$) in Sr and more radiogenic in Nd and $\varepsilon N_{d(0)}$ (Fig. 6). Differences with soil depth were significant for Sr isotope ratios reflecting greater solubility[47] relative to Nd. The ADE Sr isotope signature (0.714) falls between those of the Solimões River

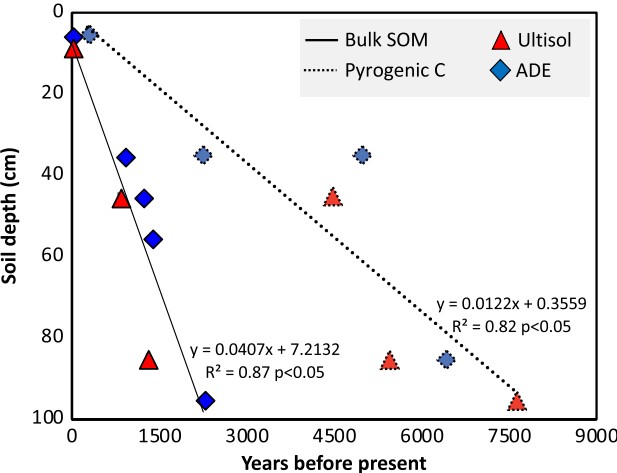

**Fig. 5 Age-depth model for bulk soil organic matter (solid line) and pyrogenic carbon (dashed line).** Each data point represents one calibrated [14]C date expressed in years before present. Propagated analytical uncertainties of the standard errors associated with each individual [14]C measurement ranged between 15 and 80 calibrated years and are smaller than symbols.

at a nearby site (0.709, constant through the year) and the Ultisols (0.7152)[48]. Similarly, $\varepsilon N_{d(0)}$ signature of the ADE falls between the values for Solimões River and its headwaters and the value for the Ultisols. This result is consistent with the geomorphic setting of the site, which indicates an alluvial contribution of nutrients to the site in the past. A continuous bluff, which we interpret as a natural levee (Fig. 1), is visible at the southwestern limits of the ADE patch. At the levee, the best estimate of terrain height is a maximum of 27 m above the modern average river water level (~10 m above the adjacent floodplain; Fig. 7). Although above flood limits today, the site is at an elevation known to have been inundated in the early Holocene and possibly into the middle Holocene[49]. We do not expect a significant contribution from the modern river water or water table because Sr and Nd are mainly found in the mineral phase. The concentration of Sr dissolved in river waters in the Amazon is in the order of parts per billion, whereas the Sr concentration in suspended sediments is on the order of parts per million[48], and Nd is often below detection limits in river water[50]. Thus, variation in Sr and Nd values were in all likelihood caused by fluvial mineral deposition that led to terrace development, which we interpret as the local manifestation of regional paleo floods identified in several other depositional sites[51–53].

Further support for the fluvial influence on the ADE soils is provided by differences in particle size distribution. Specifically, the classification of ADE profiles as sandy clay loam and adjacent Ultisols as very clayey has been tentatively attributed to the fusion of clay and organic matter into sand-sized particles by fire[5]. Our data point to a more parsimonious explanation; that is, alluvial deposition changed nutrient and particle size profiles at the site. As a first approximation, radiogenic signatures indicate that ~24% of Sr mass in ADE profiles originate from either fishbone or river sediment (Eq. 3), as both sources have the same Sr isotopic composition[54]. It is important to note that the Sr/Ca ratio in fishbone (0.004) is much lower than in river suspended sediments (0.02). Assuming a concentration of Sr in the soil of 200 mg kg$^{-1}$ and in fishbone of 400 mg kg$^{-1}$, the final Ca concentration of the mixture would be about 10 times higher than that of Sr, which is approximately what we see in our results. Similarly, the Nd mixtures indicate riverine inputs with significant differences between ADE and Ultisol, albeit with

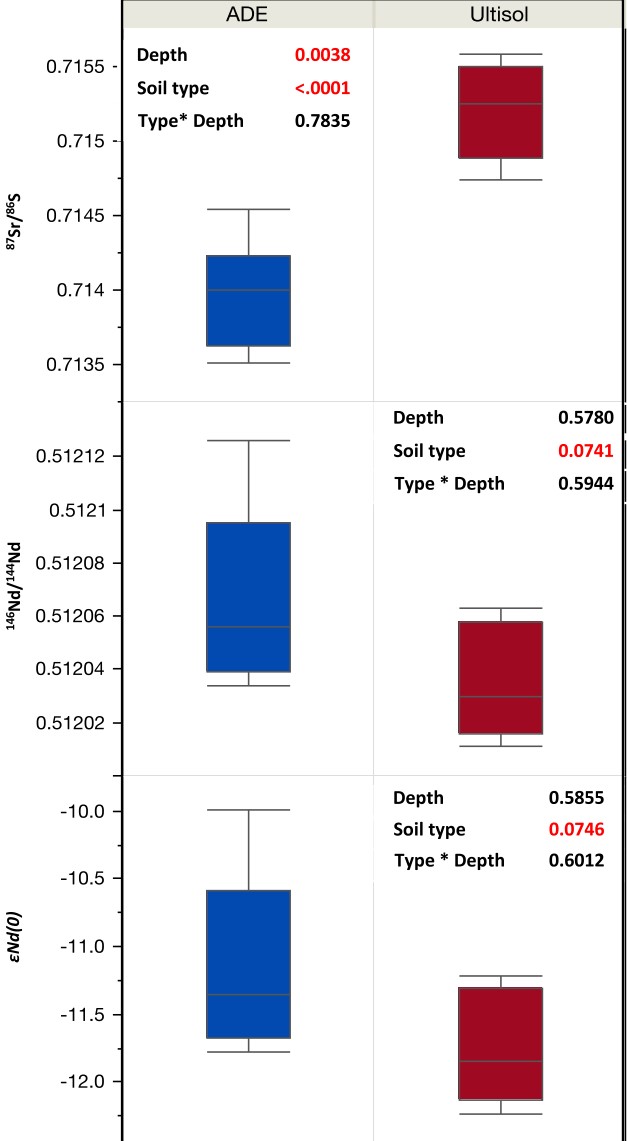

**Fig. 6 Isotope ratios of strontium (Sr), neodymium (Nd), and radiogenic epsilon neodymium ($\varepsilon$Nd (0)).** The box plots show 25th, 50th, and 75th percentiles and the error bars show the data range ($n = 8$ for ADE and $n = 5$ for Ultisol). In all cases, a two-way analysis of variance shows significant differences between soil types.

larger deviations from the mean (Fig. 6). Clay mineralogy characterisation using X-ray diffraction showed no differences in the type of dominant clays found in ADE and Ultisol profiles (Supplementary Fig. 3), but that can be explained by two factors. First, across the alluvial plain of the Solimões-Amazon River, optically stimulated luminescence and geomorphological data show large fluvial features of weathered sediment deposited at the Pleistocene-Holocene boundary[55] (long before the deposition of carbon and nutrients described above). Second, no neo-formation of minerals is expected for the highly weathered assemblages (e.g. kaolinite and iron/aluminium oxides, such as goethite) that are abundant in river sediments of this section of the Amazon basin[56]. Thus, it is not surprising that clay mineralogy is the only measured variable to converge between ADE and Ultisol.

Taken together, our findings underscore the need for a broader view of landscape evolution as a path towards understanding ADE formation and redirecting applications for sustainable land

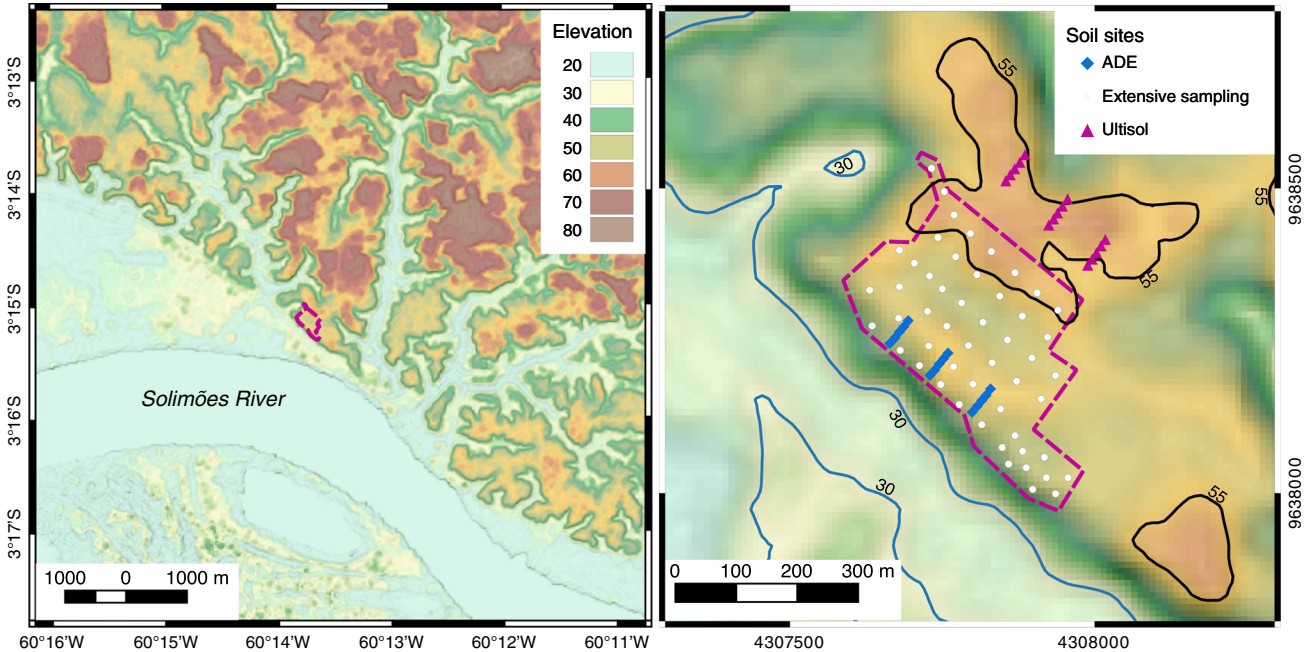

**Fig. 7 Topographic position at the study site.** Elevation from SRTM data shows the study site occurs on a common terrace (45–55m). The lower contour line occurs near areas of Entisols (Aquents or Fluvents) within the modern floodplain (below 30m) and the upper contour line occurs in areas of Ultisols (above 55m). The area between the floodplain and Ultisols contains the ADE.

use and conservation, such as those arising from biochar research. Other ADEs should be investigated for a potential alluvial origin, on the basis of physical properties and elemental sources, to improve basic knowledge of the transition from nomadic to sedentary populations in Amazonia and their influence on socioecological trajectories.

**Human inputs**. In pre-Columbian times, the most likely human-derived nutrient inputs to soil were food waste and faeces. Archaeologists hypothesise that such domestic waste deposits led to the widespread formation of anthropic soils near ancient settlements[18], in a process that some authors have described as non-intentional[57]. Molecular markers of human faeces found in several ADEs, including our site[19], offer support for that hypothesis. However, we still do not know how much waste would have been needed to form a typical ADE patch. Here, we offer a first approximation based on the excess amount of P and Ca, calculated in relation to the surrounding Ultisol at our site (~32 Mg ha$^{-1}$; Fig. 3), which implicitly represents the depositional history of many other elements that correlate strongly with those macro-nutrients (Supplementary Fig. 1).

At the high end of the nutrient-amount-per-waste-mass spectrum, more than 5300 Mg ha$^{-1}$ of fresh fish biomass would be needed to explain ADE's nutrient excess. That is, assuming that fresh fish is ~80% water and that P and Ca combined amount to 6% of the total dry mass of edible freshwater fish[19], of which we speculate no more than half would have been added to the soil. In that scenario, using 20–60 people per hectare and assuming that each adult consumed, on average, a maximum of 219 g and a minimum of 90 g of fresh fish per day[58] we calculate that an ADE patch of 1 ha would have required ~1100–8000 years of continued occupation (Eq. 4). By the same logic, at the low end of the nutrient-amount-per-waste-mass spectrum, the time necessary for ADE formation if human faeces were the sole source of nutrients would be ~3700–11,000 years. That is, assuming the same 20–60 people per hectare and average values for the faeces eliminated by each adult as 128 g containing ~80%

water and 15 mg g$^{-1}$ of P and Ca in the remaining dry mass[31], of which all could have been added to the soil. These are conservative estimates that disregard losses when, typically, a large portion of nutrients are expected to leave the soil after deposition. This is particularly the case for Ca, which, once mineralised, can be quickly removed from the soil profile by rainfall. The inherently efficient cycling of Ca in the tropical forest could lead to little loss over time, but that would change under management and would be recorded in litter and soil stoichiometry[59]. The addition of charcoal can slow down Ca loss but leaching rates of 50–150 kg Ca ha$^{-1}$ year$^{-1}$ have been reported in charcoal-rich ADE profiles[60]. Moreover, P losses are expected to be slower than for Ca, although large amounts of P can be lost in fine particulate form as a result of burning—a practice often attributed to ADE formation[61]. If we were to account for those losses, the total amount of waste needed to form the 12-ha ADE patch at our site would be orders of magnitude larger than those presented here.

A synthesis of anthropological studies from riverine areas where ADEs are typically found indicates population densities ranging from 20 to 60 people per hectare in sedentary villages in 1492 AD and suggest that at least five million people inhabited the Amazon region prior to European contact[58]. Those estimates fall within the range of dense pre-Columbian landscapes elsewhere in the Americas. For example, in 1519 AD the Aztec empire was comprised of several million people, living in city-state capitals with a median density of 50 people per hectare[62]. Population estimates can have large margins of error and should not be used to generalise patterns across the region. The Amazon basin has a complex history of occupation and land use and, prior to European contact, indigenous peoples relied on >80 different plant species associated with settlements that possibly covered 0.1% of the region[63]. A recent analysis of cultural transitions suggests that specialised land-use systems faced major reorgani-sation during Holocene climate fluctuations discussed above, whereas others that relied on polyculture agroforestry were more resilient and remained largely unaffected[64]. Our analysis of elemental amounts and sources is site-specific and it is not

intended to capture regional socioecological transitions, but the clear picture emerging from our data is that indigenous populations identified and actively used ADEs after their genesis. It is possible that dense populations occupied riverine areas uninterruptedly for thousands of years longer than previously recorded, contributing large amounts of fish waste to the soil, as it appears to have been the case for some of the first hunter–fisher–gatherer communities in western Amazonia[65]. It is also possible that those communities were capable of recycling nearly all of their waste into ADEs, with little to no loss through leaching or burning. It is more plausible, however, that human-derived inputs represent a minor fraction of ADE's chemical makeup, a fraction that, we hypothesise, was introduced in the relatively recent past. If our hypothesis holds true, other sites will exhibit similar depositional patterns and biogeochemical signatures as those reported here.

**Final considerations**. We found direct and indirect evidence that natural processes created a well-studied ADE site in the Brazilian Amazon. Our data reveal external inputs of carbon and mineral elements several thousands of years earlier, and in several orders of magnitude greater amounts, than those expected from human activity. Across the Amazon basin, three main phases of human occupation are thought to have occurred: a pre-cultivation period (>6000 years ago); an early-cultivation period (6000–2500 years ago); and a late-cultivation period (2500–500 years ago)[15]—a chronology of occupation that is supported by genomic, linguistic, and archaeological evidence[66]. At our site, the enrichment of microcharcoal and mineral elements (two critical ingredients needed for ADE formation) began at ~7630 years ago, that is, before the earliest evidence of soil management that characterizes the late-cultivation period. To achieve the observed levels of P and Ca enrichment at the site, large human populations would have had to actively manage soils continuously for thousands of years prior to the currently accepted chronology of sedentarisation. A more parsimonious explanation is that indigenous peoples used their knowledge to identify and preferentially settle areas of high fertility before the emergence of plant cultivation. This hypothesis implies that pre-Columbian societies understood and exploited natural processes of landscape formation, which led to the unique properties of ADEs, but were not responsible for their genesis. Our findings do not preclude, however, a more recent human effect on the local environment. The wisdom of native populations, manifested for example in the well-documented recent application of waste materials to the soil, may well have further enriched ADE profiles or, at least, countered their otherwise-inevitable degradation. The extent to which other ADE sites originated from natural depositional processes or were managed for long-lasting fertility remains unclear. New discoveries are to be expected from interdisciplinary research that combines indigenous knowledge with biogeochemical and geographical data to study landscape development as a path toward sustainable land use in the Amazon and other tropical environments.

## Methods
**Site description and sampling scheme**. The EMBRAPA-CPAA site covers 210 hectares of managed and unmanaged land just north of the Solimões River near its confluence with the Negro River, within the sedimentary basin of the Amazon River. Satellite and field measurements show that the terrace is located ~40 m above the modern sea level, at a maximum of ~27 m above the modern river level, or ~10 m above the adjacent floodplain (3° 15′ 6.8″ S and 60° 13′ 46.9″ W). The relative topographic positions of ADE and Ultisol at the site matches the classic bluff model of prehistoric riverine settlement (close to the river and with fresh sediment providing enhanced fertility)[67]. A bluff is indeed visible at the southwestern limits of the ADE patch, which is interpreted as a natural levee formed due to Holocene paleo river dynamics (Fig. 1). This interpretation is based on other features of the

Solimões-Amazon fluvial system, between the Negro and Madeira tributaries. Specifically, records of paleo river dynamics in many extensions of the Solimões River indicates that the combined action of neotectonics and rainforest establishment related to the increase of humidity in Amazonia ~6000 years ago allowed for sediment accumulation and asymmetric distribution of fluvial terraces throughout the region[53].

Lidar data was not available for the study area, but the comparison of NASA's SRTM and nearby ICESAT-2 satellite data show with no systematic offset, where blue colours below 30 m are within the 100-year flood level (Fig. 7). Across the slight topographic gradient found in the modern terrace at our site, we determined the numbers of transects and soil samples to be collected across using a priori test of statistical power, which anticipated two main effects: soil type and soil depth, assuming a moderate statistical power of 0.8 for each effect[68]. Thirty maximum contrast profiles were sampled with a Dutch auger every 10 cm up to 1 m depth to a total of 300 samples of ~200 g each. Uncompressed soil cores were obtained continuously at each profile using metal rings to determine the bulk density at each depth. Coarse litter was removed from the topsoil before sampling. Prior to analysis all samples were allowed to air-dry for 2 weeks under the shade, sieved to <2 mm, sterilized, and finely ground in an agate ball mill.

**Organic carbon analysis**. Total soil organic carbon was determined through combustion and gas chromatography in a standard elemental analyser (Costech, USA). For pyrogenic carbon determination, the fine soil fraction was dried and homogenised. Aliquots from each sample were ground in a ball mill and digested using acid-peroxide[69]. Ground aliquots (1 g) were placed in Erlenmeyer flasks (50 ml) to which 30% $H_2O_2$ (20 ml) and 1 M $HNO_3$ (10 ml) were added and swirled by hand at room temperature for 30 minutes. Flasks were heated in a water bath to 90 °C for 16 hours. After digestion, the samples were filtered through pre-weighed Whatman glass filters (1.21 μm), which were dried at 60 °C for over 24 hours and weighed to obtain the mass of residual material after filtration. Aliquots were oven-dried and the mass of the residue (digested sample) was determined by weight difference. Micro-charcoal particles (<150 μm) comprise most of the charcoal pool at our site and very small particles (<40 μm) dominate pretic ADE horizons[5]. Accordingly, the results obtained here using digestions of the whole fine soil fraction, is considered to be representative of the whole microcharcoal pool. The estimated amount of micro-charcoal can be calculated for each soil type assuming that all non-charcoal organics were fully consumed during peroxide-acid digestion:

$$PyC = \left(\frac{C_1 * M_1}{M_2}\right), \tag{1}$$

where $PyC$ is the mass of pyrogenic carbon; $C_1$ is the carbon concentration in the digested sample; $M_1$ is the mass of the digested sample, and $M_2$ is the mass of the original soil sample.

**Radiocarbon dating**. Soil carbon age-depth models were developed using calibrated radiocarbon ([14]C) activity of bulk soil organic matter and pyrogenic carbon from multiple depths in ADE and Ultisol profiles. Radiocarbon dating and calibration of [14]C ages to years before present were performed at the University of California Irvine, through accelerator mass spectrometry. Samples were collected in soil pits dug adjacent to each transect to determine the age of 15 samples (8 bulk soil organic carbon samples and 7 PyC samples) collected from ADE and Ultisol horizons, as described above. Bulk carbon samples were analysed after drying and sieving. A modern charcoal standard was created by combusting dry modern wood tissue from plants growing near the study. All pyrogenic carbon samples, including the modern standard and an old charcoal standard with no detectable radiocarbon activity (for contamination control purposes), were cleaned using alternating heated 10% KOH and 10% HCL rinses prior to analysis. Human activity and natural disturbance may have contributed additional charcoal into the soil matrix, and thus the calibrated [14]C dates reported here represent the most recent of a range of possible ages of carbon input.

**Elemental analysis**. Extractable P and Ca were determined using atomic absorption spectrophotometry. Mehlich 1 (0.025 N $H_2SO_4$ + 0.05 N HCl) was used for P extractions and KCl (1 N) was used for Ca extractions. For total nutrient concentrations, ~200 mg of finely ground soil was digested in PFA vessels on a Teflon-coated graphite block digestor (Model A3, Analysco Ltd, Chipping Norton, UK). Acid digestion was initially with 2 ml of $HNO_3$ (68% Primar Plus grade) and 1 ml of $HClO_4$ heated at 80 °C for 8 h and subsequently at 100 °C for 2 h. The following day 2.5 ml of HF (40% AR) was added to the tubes and the temperature raised to 120 °C for 1 h, increased and maintained at 140 °C for 3 h, then 160 °C for 4 h; the vessels were again left overnight at room temperature. On the third day, the block digester was set to 50 °C and 2.5 ml of $HNO_3$ and 2.5 ml Milli-Q water (18.2 MΩ cm; Fisher Scientific UK Ltd, Loughborough, UK) were added to tubes and left at 50 °C for one hour. After cooling, the digested samples were washed with Milli-Q water into 50 ml plastic flasks and stored in a 5% $HNO_3$ matrix at room temperature in universal sample bottles pending elemental analysis. The standard reference material (SRM 2711a Montana soil, National Institute of Standards and Technology-NIST, Gaithersburg, MD, USA) and blank digests were included with every batch of 48 samples. Subsequent analysis of Ag, Al, As, Ba, Be, Ca, Cd, Co,

Cu, Cr, Cs, Fe, K, Li, Mg, Mn, Mo, Ni, P, Pb, Rb, Se, Sr, Ti, Tl, U, V and Zn was carried out by inductively coupled plasma mass spectrometry (ICP-MS; Model iCAP-Q™, ThermoScientific, Bremen, Germany). All samples were blank-corrected using the operational digestion blanks[6].

**Isotopic analysis.** Samples from the surface and deep soil layers were used to determine stable isotope ratios of carbon, strontium, and neodymium. For carbon isotope ratios ($\delta^{13}C$), bulk organic carbon was determined by dry combustion gas chromatography coupled with continuous-flow isotopic-ratio mass spectrometry (GC-IRMS; Europa, Crewe, UK) at the University of California, Davis. To calculate the contribution of forest and savannas at different points in time, the $^{14}C$ age-depth model was combined with typical $\delta^{13}C$ signatures of $C_3$ trees ($-25.9‰ \pm 0.4$) and $C_4$ grasses ($-11.2‰ \pm 0.2$) in the study region[43] in a two-end member mixing model[70]:

$$C_4 = C_t(\delta_t - \delta_3)/(\delta_4 - \delta_3), \tag{2}$$

where $C_t$ is the total mass of organic carbon in the soil coming from $C_3$ and $C_4$ sources; $C_3$ is the mass of carbon derived from trees and herbaceous plants that assimilate carbon through the $C_3$ metabolic pathway; $C_4$ is the amount of carbon derived from grasses which assimilate carbon through the $C_4$ metabolic pathway; $\delta_t$ is the $\delta^{13}C$ ratio of $C_t$; $\delta_3$ is the $\delta^{13}C$ ratio of $C_3$-derived organic carbon, and $\delta_4$ is the $\delta^{13}C$ ratio of grass-derived organic carbon expressed as the relative (percentage) of grass input to the total organic carbon pool.

The isotopic ratios of soil carbon incorporate changes in atmospheric $CO_2$ due to the emission of fossil fuels over the past century. However, changes in $\delta^{13}C$ ratios of $CO_2$ have been small for most of the Holocene (<1.5‰ over the past 10,000 years[71]) and affected both tree and grass signatures. We can therefore rule out this effect as an explanation for long-term changes in carbon source observed here. After organic matter deposition, diagenetic fractionation occurs primarily in the topsoil as a function of decomposition, typically leading to a <2‰ fractionation[72], a result that can be observed here within the top 20 cm of ADE and Ultisol profiles (Fig. 4). Other sources of variation, such as changes in plant water-use efficiency in response to atmospheric $CO_2$, are not relevant at the scale studied here because this effect would be smaller than differences between $\delta_3$ and $\delta_4$. We can also rule out any confounding effect of inorganic carbon, which were absent from our samples, as expected due to moderate acidity (pH 5.11–5.73) of ADEs and extreme acidity of Ultisols at our site (pH 4.12)[5].

For the analysis of Sr and Nd isotopes, soil samples were digested in two stages using concentrated HF/HNO$_3$ and 6 N HCl in Teflon Savillex beakers placed on a hot plate. In order to determine the $^{87}Sr/^{86}Sr$ ratio, another aliquot of the initial solution was submitted to chromatography over SR-B50-A resin (Eichrom, Lisle, IL, USA; particle size 100e150 m) using 2.9 M HNO$_3$ as eluent. The determination of $^{87}Sr/^{86}Sr$ ratios was performed using a ThermoFisher Triton, involving 30 cycles in one block. During the course of sample measurements, repeated determination of Sr isotopes in a 100 ppm solution of NIST SRM 987 strontium carbonate ($^{87}Sr/^{86}Sr = 0.71036 \pm 0.00026$), yielded an average value of $0.71025 \pm 0.00008$, $n = 12$. An aliquot of this solution was used for mineral extractions adapted specifically for Amazonian Oxisols, in which trace elements were separated as a group using cation-exchange columns[73]. Following this procedure, a reversed-phase chromatography was applied for Samarium–neodymium (Sm–Nd) radiometric dating using columns loaded with HDEHP (di-2-ethylhexeyl phosphoric acid) supported on Teflon powder. A mixed $^{149}Sm$–$^{150}Nd$ spike was used with re-evaporation filaments of a double filament assembly. The isotopic analyses were performed using a ThermoFisher Triton multi-collector mass spectrometer in static mode. Uncertainties for Sm/Nd and $^{143}Nd/^{144}Nd$ ratios were ±0.1% (2σ) and 0.005% (2σ), respectively, based on repeated analyses of international rock standards BCR-1 and BHVO-1, normalised to a $^{146}Nd/^{144}Nd$ ratio of 0.7129 using blanks smaller than 100 pg. Considering that ADE is a mixture of weathered minerals found in typical Ultisols with exogenous inputs (Supplementary Fig. 3), the contribution of river sediment to the total pool found in the ADE was calculated using the mass of Sr and its stable isotope ratios ($^{87}Sr/^{86}Sr$) measured in the surrounding Ultisol and river water as follows:

$$Sr_{ADE} = Sr_{Ultisol}\left(^{87}Sr/^{86}Sr\right)_{Ultisol} + Sr_{river}\left(^{87}Sr/^{86}Sr\right)_{river}. \tag{3}$$

*Population density estimates.* The first step we used to estimate the size of human populations and the amount of inputs needed to create the studied ADE was to calculate how much additional mass of human inputs would be needed to explain excess Ca and P in the ADE relative to the stocks in the surrounding Ultisol (for a total soil depth of 1 m; Fig. 3). By focusing on Ca and P stocks we implicitly represent the depositional history of 16 other mineral elements which correlate significantly with P and Ca at the site (Supplementary Fig. 1). The second step to calculate human inputs was to consider the possible anthropic sources of Ca and P under the assumption that population densities are directly related to the amount of waste deposited onto soils. To minimise uncertainties related to type of biomass deposited we considered three sources that are often attributed to ADE genesis: human faeces, plant biomass, and fish biomass. Fish waste is the richest of those sources, and thus, it yields the most conservative population densities. The third step was to consider the time needed to generate the observed excess fertility in the

ADE via the consumption of fish by a certain number of people:

$$Q = D_t\,0.2M_t/TP, \tag{4}$$

where $Q$ is protein available from fish per capita per day in grams; $D_t$ is the number of person-days fishing for period $T$ in days; $M_t$ is the dressed weight in grams of the average day's catch of one for the period, and $P$ is the average population subsisting on the protein obtained over $T$. The constant (0.2) is a rough value indicating the protein yield from any given quantity of fresh, dressed meat or fish obtained by inspecting the values for protein in 100 g edible portions of several kinds of meat and fish in indigenous communities of Latin America[58].

**Reporting summary**. Further information on research design is available in the Nature Research Reporting Summary linked to this article.

## Data availability
The data used to produce all figures and supplementary materials present here are available at https://doi.org/10.7264/9qdm-en61

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

## Acknowledgements

We thank the National Science Foundation (AGS#1602958; Convergence Accelerator #1939511), the Brazilian Council for Scientific and Technological Development – CNPq, and the 2019 Resilience Initiative Interdisciplinary Award at the University of Oregon for research funding. All samples were collected, processed, and analysed in coordination with the local authorities and following Brazilian laws and regulations. We thank Dr William Horwath for logistical support in the USA and Dr Wenceslau Geraldes Teixeira for providing logistical support in Brazil and for hosting international training events including field courses at EMBRAPA CPAA, which catalysed this collaboration. The publication fees for this article were supported by the University of Oregon Libraries Open Access Article Processing Charge Fund.

## Author contributions

L.C.R.S.: Designed the study, generated and analysed data, made figures and wrote the manuscript. R.S.C.: Designed the study, collected samples, interpreted elemental data, and revised manuscript. J.L.W.: Integrated data sets and performed spatial analysis. B.B.: Analysed charcoal samples and interpret biogeochemical data. L.H.: Digested samples and quantified pyrogenic carbon. D.G.: Generated elevation maps, oversaw pyrogenic carbon analysis, and revised manuscript. A.W.M.: Collected soil samples, provided

resources, and preliminary data. G.C.M.: Collected soil samples and generated preliminary data. A.C.V.M.: Performed laboratory work and helped with data analysis. J.Z.M.: Organised data and helped interpret elemental inputs. V.d.F.M.: Helped with the analysis and interpretation of elemental data. S.D.Y.: Provided resources and oversaw chemical analysis. M.R.B.: Provided resources, helped with multi-proxy data interpretation, and revised manuscript. R.V.S.: Generated isotopic data, interpreted geological sources, and revised manuscript.

## Competing interests

The authors declare no competing interests.
