## [Peer Review File · Nature Communications]

Reviewers' comments:

Reviewer #1 (Remarks to the Author):

"A new hypothesis for the origin of Amazonian Dark Earths" by Silva et al. reexamines the genesis of Amazonian Dark Earths (ADE) soils from ~4.5 ha study area from the Brazilian Agroforestry Research Station (EMBRAPA). Nutrients concentrations of phosphorus (P), calcium (Ca), total soil carbon, and pyrogenic microcharcoal, along with 26 other mineral elements, including strontium (Sr) and neodymium (Nd) were analysed in ADE soils and compared with nearby Ultisols (natural soils). Based on the levels of P and Ca, the authors then calculate a pre-European population density estimate needed to attain the observed nutrient enrichment in ADE fertility, relative to surrounding soils. They conclude that to achieve the accumulation of nutrients observed at the site, indigenous populations would have had to settle the site thousands of years earlier (around ~7500 BP), or be orders of magnitude larger than previously reported (ca. 20 to 60 people per hectare). Based on these calculations, they propose an alternative hypothesis that ancient populations exploited natural ADEs (e.g. riverine sediments, enriched in P, Ca, and enriched concentrations of radiogenic strontium (Sr), neodymium (Nd), $\delta^{13}\text{C}$, and pyrogenic carbon), but were not responsible for their genesis.

To date, experimental biochar studies have fallen short of replicating the long-lasting ADE fertility and the formation processes of ADEs are still unknown. Silva et al. attempt to address this important research question, however, their arguments to support the natural genesis of ADEs has fundamental flaws based primarily on the oversight of a significant body of archaeological, archaeobotanical, and palaeoecological literature that establishes the antiquity and density of local human occupation. The authors cite the eastern Amazon, Santarém region as their reference for human chronologies, using the apex of pre-Columbian activity (starting 4,500 and peaking at 2,500 BP). As a correction to the data used in this study, in the eastern Amazon, the earliest evidence of human activity is ~11,000 cal BP from Pedra Pintada Cave, Monte Alegre, Pará¹. There is extensive archaeological evidence led by Eduardo Neves that is closer to the study region that would be more relevant than the eastern Amazon case study. There is a high density of archaeological sites near the EMBRAPA study area (>5 archaeological sites less than 10 km away and more than 20 sites within a 30 km radius)². These records are essential to the interpretation of the ADE histories in the area as the earliest evidence of local human activity is >9,000 years ago 2–6 and in the broader region >11,000 years ago^{1,7,8}. If the accurate chronology of human occupation in the region is considered, it is not possible to exclude humans as the dominant driver of ADE formation.

In addition to the archaeological data, there have been decades of extensive debate on population estimates in the Amazon ranging from 1 to 10 million people in the Amazon Basin. See Koch et al. 2018 for a review of this debate and a synthesis of the key literature. Population density estimates have large margins of error and can lead to over-generalizations about the Amazon which has extremely heterogeneous human occupation histories. These types of estimates should be treated with extreme caution.

That being said, based on the calculations proposed here: an ADE patch of 1 hectare would have required ~1000-8000 years of continued occupation to attain the P and Ca observed in the record if only fish was added and ~1500 to 18000 years if only human waste was added. Based on the archaeological data from the eastern Amazon, Silva et al. argue that the apex of human activity in the region does not occur until 2500 to 500 cal yr BP (Line 84 to 86). This leads the authors to conclude natural versus human genesis of ADEs. When the accurate chronology of local human occupation is considered, the history of human occupation falls within the population density estimates produced in the study. Existing research of ADEs^{9,10} indicates a combination of both food and human waste while the results here also suggest riverine sediment input. A calculation combining all three of these inputs would be more accurate as well as decrease the time needed to attain nutrient enrichment levels. The authors argue that their calculations do not support the traditional view that ADEs were formed after a

few centuries of intermittent management. However, there are dated ADE soils <550 years old^{11,12}, suggesting that soils in densely occupied regions, could have formed in a few hundred years. Again calculations that include a combination of food and human waste and riverine sediments may improve these estimates.

In addition to the archaeological research, there is an extensive body of archaeobotanical research linking ADE and early crop cultivation throughout the Amazon lowlands >6,500 years ago^{13–16} and new studies indicate pre-ADE crop cultivation > 10,000 years ago in the Llanos de Moxos^{7,17}. Together, these data confirm ADEs were formed after the adoption of crop cultivation suggesting ADEs may have been developed by humans and used to enhance subsistence yields. Based on the inclusion of this archaeological and archaeobotanical literature, a revised interpretation of these data must be considered that includes humans and a potential driver of ADE genesis.

Another line of evidence used to support riverine inputs is from the isotopic enrichment of ADE carbon profiles that coincides with pyrogenic carbon inputs. Based on the revised archaeological chronology discussed above, this will co-occur with human occupation in the area. While the enrichment of $\delta^{13}\text{C}$ values could indeed be linked to the expansion of the C₄-dominated “campinaranas”, the presence of humans at the study site, necessitates the consideration that crop cultivation may have influenced the $\delta^{13}\text{C}$ values both indirectly (through the removal of trees for crop cultivation which increased C₄ grasses) and directly (through planting C₄ crops such as maize). Previous research has shown that pre-Columbian populations used open landscapes for crop cultivation because it was less labour intensive than clearing forested landscapes¹⁸. Additionally, maize cultivation is often preceded by a peak in sedimentary charcoal¹⁶. There are numerous examples of concurrent maize cultivation and fire management in the region dating to the genesis of the ADE soils at the EMBRAPA study site^{15,16,19}. One of the most accurate ways to determine the drivers of $\delta^{13}\text{C}$ values is through phytoliths from the ADE soils which can distinguish C₄ grasses versus maize. While the addition of phytolith analysis may not be feasible in this study, without them, it is not possible to determine if $\delta^{13}\text{C}$ enrichment was driven by C₄ grass or maize cultivation. Lastly, see Arienzo et al. 2019²⁰, who demonstrate local fire management versus regional fire activity, drove sedimentary black carbon signals during the pre-Columbian period.

The geochemical evidence produced by Silva et al. including the elevated concentrations of Sr and Nb provide compelling evidence riverine sediments were incorporated (either naturally or intentionally) to enrich the ADE soil fertility and would account for the presence of rare elements in the soils. The addition of riverine sediments may have played a key role in soil genesis by enhancing soil nutrient levels (along with fishbone and human feces). Based on the current data presented here, it is not possible to disentangle whether the riverine sediments were deposited naturally or added intentionally by humans to ameliorate soil nutrient levels, it does provide potential insights into the genesis and lasting fertility of ADEs.

The inclusion of existing archaeological, archaeobotanical, and palaeoecological data discussed above, has the potential to shed new light on the interpretations of the EMBRAPA ADE record. The data presented by Silva et al. provide some of the earliest evidence of ADE soils which adds to a growing body of evidence of early Holocene sedentarisation and crop cultivation in the region^{13,14}. I encourage the authors to consider these comments which will improve the interpretation of their results which has the potential to provide new insights into early soil amelioration techniques particularly in the realm of riverine nutrient enrichment.

References:

1. Roosevelt, A. C. et al. Paleoindian cave dwellers in the Amazon: the peopling of the Americas. *Science* (80-.). 272, 373–384 (1996).
2. da Silva Costa, F. W. *Arqueologia das campinaranas do baixo Rio negro: Em busca dos pré-ceramistas nos areas da Amazônia central.* (Universidade de São Paulo, 2009).

3. Zuse, S. Variabilidade cerâmica e diversidade cultural no alto rio Madeira, Rondônia. (University of São Paulo, 2014).
4. Caldarelli, S. B. & Kipnis, R. A ocupação pré-colonial da Bacia do Rio Madeira: novos dados e problemáticas associadas. *Especiaria - Cad Ciências Humanas* 17, 229–89 (2017).
5. Mongeló, G. Z. O Formativo e os modos de produção: ocupações pré-ceramistas no alto Rio Madeira, RO. (University of São Paulo, 2015).
6. Lima, M. D. N. O rio Unini na Arqueologia do baixo rio Negro, Amazonas. Dissertação. (Universidade de São Paulo).
7. Capriles, J. M. et al. Persistent Early to Middle Holocene tropical foraging in southwestern Amazonia. *Sci. Adv.* 5, eaav5449 (2019).
8. Lombardo, U. et al. Holocene land cover change in south-western Amazonia inferred from paleoflood archives. *Glob. Planet. Change* 174, 105–114 (2019).
9. *Amazonian Dark Earths* Dordrecht: (Kluwer Academics., 2003).
10. Woods, W. I. et al. Amazonian dark earths: Wim Sombroek's vision. (Springer, 2009).
11. Maezumi, S. Y. et al. The legacy of 4,500 years of polyculture agroforestry in the eastern Amazon. *Nat. Plants* (2018) doi:<https://doi.org/10.1038/s41477-018-0205-y>.
12. de Souza, J. G. et al. Climate change and cultural resilience in late pre-Columbian Amazonia. *Nat. Ecol. Evol.* 3, 1007–1017 (2019).
13. Hilbert, L. et al. Evidence for mid-Holocene rice domestication in the Americas. *Nat. Ecol. Evol.* 1, 1693–1698 (2017).
14. Watling, J. et al. Direct archaeological evidence for Southwestern Amazonia as an early plant domestication and food production centre. *PLoS One* 13, e0199868 (2018).
15. Brugger, S. O. et al. Long-term man–environment interactions in the Bolivian Amazon: 8000 years of vegetation dynamics. *Quat. Sci. Rev.* 132, 114–128 (2016).
16. Bush, M. B. et al. A 6900-year history of landscape modification by humans in lowland Amazonia. *Quat. Sci. Rev.* 141, 52–64 (2016).
17. Lombardo, U., Denier, S., May, J.-H., Rodrigues, L. & Veit, H. Human–environment interactions in pre-Columbian Amazonia: The case of the Llanos de Moxos, Bolivia. *Quat. Int.* 312, 109–119 (2013).
18. Carson, J. F. et al. Pre-Columbian land use in the ring-ditch region of the Bolivian Amazon. *The Holocene* 25, 1285–1300 (2015).
19. Kistler, L. et al. Multiproxy evidence highlights a complex evolutionary legacy of maize in South America. *Science* (80-.). 362, 1309 LP – 1313 (2018).
20. Arienzo, M. M., Maezumi, S. Y., Chellman, N. J. & Iriarte, J. Pre-Columbian Fire Management Linked to Refractory Black Carbon Emissions in the Amazon. *Fire* vol. 2 (2019).

Reviewer #2 (Remarks to the Author):

Review of NCOMMS-20-12744-T

This paper provides a detailed examination of the differences in a wide suite of elemental and isotope characteristics between a classic 'Amazonian Dark Earth' – ADE and the adjacent ultisol. This comparison, along with the construction of a detailed nutrient budget leads the authors to conclude that natural processes led to the enrichments observed in the ADE site rather than the systematic improvement of the soils by humans over a long period. An additional line of evidence is that the formation of the site predates known occupation of the area.

This is an excellent piece of work, very detailed and clearly presented to the point where I could not identify a typographical or grammatical error to correct (which is very unusual). IN a nutshell, I believe that the authors have made a solid case that natural processes have contributed to site formation. However, as a disinterested observer of a very interesting topic, I feel the paper currently

downplays human agency to a degree that is not supported by the data. I think the paper does warrant publication as an important contribution to an important debate, but needs some reworking to tone down the over-writing of humans as important in the genesis of ADEs.

I am convinced by the arguments that there is an exogenous input to the nutrients at the site – that is clear from the data presented. The arguments about a smaller human imprint are weaker.

(i) There is a single paper from 2003 (line 245) referred to as evidence that populations were not sufficient to generate the nutrient inputs calculated. There has been a lot of work more recently that has demonstrated considerably more evidence for widespread occupation of the Amazon basin, so I am unconvinced the population densities weren't higher than proposed in that 2003 paper.

(ii) The authors claim there is no evidence for occupation early enough to explain the radiocarbon dates at depth in the ADE. Lombardo et al. (2020) Early Holocene crop cultivation and landscape modification in Amazonia. Nature, recently suggest dates around 10,000 years ago for cultivation. This is admittedly in the western Amazon, but I think absence of evidence does not equate to evidence of absence in this case. If we accept 10,000 years then this is within the range of plausible nutrient additions calculated by the authors (line 228)

(iii) I think the evidence of C4 biomass in in TOC could equally be taken as evidence of human modification of the local landscape, it is not diagnostic either way.

(iv) Its had to tell from the way the data is presented, but it appears that charcoal concentration increases fairly dramatically up the profile – this is consistent with an increasing intensity of use as the surface is built up, with the trend in radiocarbon dates suggest there has been aggradation. Given the resistance of charcoal to degradation I would expect similar concentrations down the profile if this was due to natural processes? The same would not be true for TOC which degrades over time

Other points

(i) Right until the end, I thought 'exogenous inputs' meant nutrients from seasonal flooding, which does make sense to me, but then I learn in the methods that the surface is 40m above the river level. I suspect the annual flood pulse is not that high (maybe 10m?) and rates of uplift/incision are fairly low, so I am left unsure how the excess nutrients are thought to have accumulated. This needs to be clarified – is it excess nutrients from some time in the early Quaternary when the site was an active seasonally inundated feature, in which case I am unsure why extra Ca and P would not have had time to leach away a long time before the Holocene (since the authors claim Ca leaching can be rapid at line 238).

(ii) Round line 238 – this suggests that Ca should be rapidly removed, so will not accumulate and cites evidence for observed Ca leaching from ADEs. This I think could equally be interpreted as close cycling of Ca during a previous period of intense management, with little loss, changing following abandonment to losses for Ca under conditions of no management. In term of P, I assume the reference is to loss as fine particulates, but again, depends on how it was burnt, ie management, and losses may not have been high from this mechanism.

In summary, I think the authors have made a very valuable contribution in pointing out that ADEs have developed on 'inherently more favourable' sites, and this is worthy of publication, but there is no strong reason at this point to discount humans as having played a major role in developing them into the soils we see today.

Reviewer #3 (Remarks to the Author):

Amazon Dark Earth soils are enigmatic; they are dark nutrient-rich patches of soil existing within a

broad matrix on redish to buff colored nutrient poor soils. They have been assumed to be formed by native peoples either through the purposeless addition of household waste or by the purposeful addition of charcoal through biomass burning (along with waste additions). They are associated with archeological deposits and have attracted a lot of work and comment but have seldom been well characterized by soil stratigraphers and soil geochemists. Nor have they been well studied by soil geomorphologists. This manuscript argues that rather than native people control on ADE development, the ADEs formed by natural fluvial/pedogenic processes and were exploited later by people. The paper exploits a good set of paired comparisons between ADE soils and nearby Ultisols. There is much merit to the research.

In some cases, the ms seems to draw conclusions not merited by the data and/or uses the wrong data to evaluate a point. For instance, although the ADE and comparison Ultisols are close together it is not immediately obvious whether they are derived from different alluvial deposits separated in time. The ms argues that indeed they are the same deposit and uses the similarity in clay mineralogy (not elaborated) to argue that point. In fact clay mineralogy is the most likely component of the soils to converge to similarity over time and so it would be much more useful to use sand mineralogy to look for similarity/difference. Much of the interpretation depends on 13C and 14C data. In the case of the former the difference between the ADEs and Ultisol values is too small to be reliable given the potential differences in water use efficiency of different plants and the slight shift toward more heavier C in the ADE is interpreted to mean that savanna grasses grew on the site in the past. Seemingly neglecting the possibility of OM contribution from corn grown on the site. The mean residence time 14C values for charcoal in the ADEs and the Ultisols are similar, although charcoal is in greater abundance in the ADE. The explanation for the similarity and difference between the between the two soils and their underlying stratigraphy depends on the use of Sr and Nd isotopes to draw linkages between the ADE isotopic signatures and that of river sediment and then there is an unsatisfying linkage drawn with sites much further upstream. This explanation is key to the paper and is not at all well explained. Without it the rest of the ms is enticing but empty. We are asked to take this site as iconic for many other ADEs but the linkage is not made – I suspect it is there and lies in the location of the ADEs along bluffs at the edges of rivers but even the context of the site under consideration to the existing nearby river is not presented (how far above flood level is the site? What is the down cutting rate of the river? How close to the river flood level could the river have been at the time of hypothesized inundation? Or is it an ongoing process? Why focus on superposition of elements via floodwaters as opposed to imposition of a different fluvial strata underlying the ADE?).

Below I have specific comments.

L. 57-87. One of the key points about most ADEs is that they are spatially associated with river channels and as Denevan suggests may be even more associated with "bluff" habitations. This is an essential part of the story being presented here, if I understand right. The argument being laid out is that the ADEs have somehow been augmented by natural processes likely associated with fluvial processes (more on this below). It seems like it would make sense to start developing the spatial association with rivers and bluffs up front in these paragraphs.

L. 78-84. There are aspects of this sentence that don't match with the evidence laid out by Macedo et al (2010) that the soils in western Amazonia are less weathered and higher in nutrients than those in central and eastern Amazonia. There is a concern here with the sweeping nature of some of the assertions in this paragraph. The idea presented here is that the soils everywhere are nutrient poor even though the broad evidence is more complex. Denevan was obviously working in a data-poor world with regard to soil properties. I don't have a problem with the essence of the paragraph, but the building blocks have been mis-represented to some extent.

L. 104. Having trees there that are typically ~12 m high suggests that they must be regrowing secondary forest rather than relatively stable forest communities, which makes sense if it is near an area that humans have utilized previously but a bit more nuance to the statement about dense

rainforest with 12 m tall trees would be useful here.

L. 109-113 and Figs. 1 and 2. The use of the term "exchangeable" with respect to P and Ca is incorrect usage if the ions in question if a Mehlich-3 extractant was used. There is no measure for P that is considered exchangeable and for Ca exchangeable would be expected to be determined by one of several standard methods such as NH₄Ac or BaCl. And I don't understand the presentation of Ca as cmolc/kg if cation exchange was not measured (at least there is no mention of it). If Ca was measured on the Mehlich-3 extract, then it too should be presented as mg/kg in analogy with P and both should be called "extractable" rather than exchangeable. Fig 2 calls them "available" which is fine although it should be mentioned somewhere that you are assuming the Mehlich-3 extract is a measure of plant available ions. Also the last column of Fig. 2 has a funny ratio that should be rectified – if it turns out that Ca was actually derived from a measure of exchangeable extraction and therefore presented correctly in the second column then the values presented here should be transformed from molar to mass units prior to developing the ratio (remembering that what is being presented in the second column is the molar amount of charge that Ca is providing rather than a mass unit). Regardless the extraction technique(s) should be explained in the methods description rather than just mentioned in the fig caption. Fig 1 caption calls out Bottom Left and then describes the central left figure (which has been used in a previous publication and is fine but should be noted as such).

L. 132. In several places the term "rare mineral elements" is used which is confusing. I don't know whether to interpret that as another way of saying rare earth elements which has a specific geological meaning or whether it is being used to denote all elements that are typically found in low abundance in crustal rocks and soils. If the latter I would suggest using trace element as a term and if the former to use rare earth element.

L 140-148. A couple of points and questions. Given the list of 16 elements can you say anything about those that might be biocycled and those not? In passing, Se is a potentially interesting tracer because it is not much utilized by plants but required by animals and hence its distribution could be indicative of fish parts or not. It would be good to actually note what the clay minerals were. And frankly it is not surprising that the clay minerals are not different given that the formation of those minerals is guided to a great extent by climate with their amount being determined by time (or inheritance from parent material). So It would be much more instructive to quantify the sand or silt mineralogy to see if there were differences that might be indicative of either different parent materials or different lengths of weathering and hence potentially greater depletion of weatherable minerals in the Ultisols relative to the ADEs where the argument if I understand it right is that there has been augmentation.

L. 152-183. There is a lot of important material in this paragraph, but I cannot fully grasp it. I get that the black carbon dates older than the bulk carbon. This makes sense if we assume that black carbon is a more recalcitrant fraction of the total. But one needs to be careful in the interpretation since all ¹⁴C values in soil are going to be mean residence time values. I don't think there is a problem with the interpretation that the black C is older than when it is thought that people were there but is it not possible that it just means people were there earlier than we thought. After all the age of the peopling of the Americas keeps getting pushed older. That should be considered as possibility here even if you want to argue that the weight of evidence for the ADE formation lies elsewhere.

The fact that the ADE profiles contain more organic matter derived from C₄ plants is significant to the story but unfortunately the ¹³C values in the ADE and Ultisol are too similar to be interpretable as a mixing of C₃ and C₄ plants – it could easily be due to differences in water use efficiency driven by the different water holding capacities of the soil. But if you believe that you can pick apart a plant signature between del ¹³C of -24 and -26: Was that C₄ contribution derived from savanna grass or from corn grown in the ADE area? The interpretation put forth here relies on work showing that elsewhere in Amazonia there are/were patches of savanna that contributed C₄ signature to soils. That discussion requires more explication than is provided here – really needs its own paragraph to lay out the argument because it seems as though the simplest explanation is the corn crop contribution – so

why make it more complicated? Is it really legitimate to point to savanna sites farther upstream where it is more likely that most of the soils are more nutrient rich (at least according to the papers by Quesada et al)? I know these savanna patches and their potential significance to past conditions have been deemed important to the paleoecological relationships in the basin but how clear is the relationship? The relationship between upstream patches and the ADE site under consideration here is key to the argument but not well enough explicated.

The material from L. 165-177 is presented as an explication of the savanna story and related ideas about depositional environments but it is too compact to provide the context necessary to understand the problem. For instance, I do not understand the quote about sediment deposition being activated or deactivated by specific environments. What landscape positions are being talked about? Is this in the floodplain (Varga forest) or in the upland (Terra Firmae forest)? And since there has not been an explicit discussion about the landscape position of the study location I can't envision the process being called on. From other reading (Denevan, Glaser, etc) I assume that we are talking about bluff sites that are in close proximity to the floodplain but I don't know that to be the case. Is the paleochannel being discussed here envisioned as contemporaneous with the process dynamics going on in the floodplain or is it etched into the upland? And what is the relevant time frame? – We know that the soils have different clay contents, is that because of different ages between the "levee" and the Ultisol site? An understanding of the stratigraphy at the contact between the two different landforms would be helpful here. I guess maybe the problem here is that the results are mixed with the discussion and the latter is not well enough developed.

L. 209-211. In Fig 6; left panel, what do the notations inside the graphs mean? And the Sr isotope Y-axis is screwed up. Somehow the graphic on the right side seems to have a lot of extraneous information that is beside the point – but maybe that is just because the depositional model the ms wants to get across is not well enough presented to include that material? Within reason I find the Sr and Nd isotope data to be convincing that there is a riverine input and that perhaps it is better to call on direct input of sediment than on the importation of fish bones although the Sr/Ca ratios could be jacked by thousands of years of differential leaching. Although it looks like there might not be much statistical power for Nd. Still I am having a hard time imagining this fluvial input being superimposed on the Ultisol soil/landscape rather than having two different sedimentary deposits separated by enough time to allow the Ultisol to have evolved to its present state of clay concentration whereas the "levee" deposit has evolved from a relatively sandy deposit to a sandy clay one. A focus on clay content rather than clay mineralogy. What is the particle size of the sediment that has supposedly been deposited on the levee (that derived from the suspended sediment load in the river? How does this augmentation actually work? What are the relative topographic positions of the Ultisol and ADE? What does the mineralogy of the sand or silt fraction tell you about sedimentation? Too many questions here to allow an unambiguous interpretation of the isotope data. Also not to be too argumentative or fanciful here but what about Denevan's bluff model of habitation assertion that the location of habitation had to balance several features of the landscape: need to be close to the river and need to be close to fresher sediment that could be carried to the ADE sites for mixing with the existing soils to enhance nutrient status. The latter is perhaps more of an Anthropological construct than reality, but I do think that the isotope data and particle-size data could be simpatico with that model as well. And that plays into the first sentence in the following Human Inputs section. Are you implicitly discounting the model that humans added fresher sediment to the ADEs? If the argument here is that natural conditions drive the difference between ADE and Ultisol then that model probably needs to be addressed head on.

L. 244-245. Is this a fair summary of the ideas (several centuries of ADE formation) in the literature about longevity and size of habitation? – seems like there is a lot of debate about this. For instance, Maezumi et al 2018 put the development of ADE technology at ~2500 years ago (at least in eastern Amazon). Perhaps a better way to discuss the results would be in terms of how they shed light on the existing ideas along with the idea that the ADEs are pre-existing features that were utilized by people.

L. 264-266. The model proposed here for the "natural process" control and development of ADEs is not compelling. It likely due to too short presentation. Need to carefully lay out the argument. And it might be better to go to more of a results section and a discussion section format to drive that communication.

Oliver Chadwick

Response to Reviewer 1

Overview: The reviewer stated that our data "provide compelling evidence riverine sediments were incorporated to enrich the ADE soil fertility and would account for the presence of rare elements ... which has the potential to provide new insights into early soil amelioration techniques" and suggested that our results "provide some of the earliest evidence of ADE soils which adds to a growing body of evidence of early Holocene sedentarisation and crop cultivation in the region". On the other hand, the reviewer identified several significant areas in need of improvement, which can be organized around two main themes: **(i)** population density estimates and **(ii)** chronology of occupation.

We appreciate this reviewer's constructive criticism of our work. In this response letter, we focus on these two major concerns, noting that many other improvements were made in the revised manuscript to address the reviewer's comments (e.g. revisions in figures and text as well as the citation of several new references). All major and minor changes are shown in the enclosed annotated PDF.

(i) Reviewer 1 argued that population density estimates have large margins of error and can lead to over-generalizations about the Amazon which has extremely heterogeneous human occupation histories.

We agree with the reviewer and we have added language to emphasize that these types of population density estimates should be treated with caution. For example, in the "Human Input" section we explain that "It is important to note that local population density estimates have large margins of error and should not be used to generalize patterns across the Amazon basin, which has heterogeneous histories of human occupation and impact". To minimize this uncertainty, we chose to use the most conservative estimates of human input, which do not account for nutrient losses after deposition of inputs needed to meet ADE's excess nutrient budget. Indeed, as explained in the original submission, if we were to account for those losses, either the total number of people per area or the total time of continuous occupation would have had to be orders of magnitude larger than the values reported here.

The reviewer also argued that "existing research of ADEs 9,10 indicates a combination of both food and human waste while the results here also suggest riverine sediment input. A calculation combining all three of these inputs would be more accurate as well as decrease the time needed to attain nutrient enrichment levels".

We agree with the reviewer about the presence of other types of waste input and we did cite previous findings to support that claim. However, the second part of the reviewer's argument is incorrect for three reasons:

First, we did calculate the amount of feces and sediment needed to attain ADE fertility. Both river sediment and human feces have more than one order of magnitude less nutrients on a mass basis than fishbone. Thus, fishbone provides the most conservative estimate of population densities and time needed to attain the observed enrichment levels.

Second, the reviewer appears to have overlooked a detailed explanation (and references cited) that we presented in the discussion. Here is a short fragment of the original text: "In pre-Columbian times, the most likely human-derived nutrient inputs to soil were food waste and feces⁵. At the high end of the nutrient-amount-per-waste-mass spectrum, fishbones are the richest source of P and Ca available as a soil amendment. At the low end of the spectrum, human faeces have significantly smaller nutrient concentrations, but were a consistent source of input to the soil, as indicated by molecular biomarkers⁴³ ... we calculate that an ADE patch of 1 hectare would have required ~1000-8000 years of continued occupation [using fishbone as the sole source] ... By the same logic, if human faeces were the sole source of nutrients, the time necessary for ADE formation would be ~1500 to 18000 years".

Third, as an acknowledgment of uncertainties in population density estimates we explain at the end of the discussion that it is possible that dense populations occupying riverine areas for thousands of years were capable of recycling all of their waste into ADEs. It is also possible that inherent differences in rates of plant uptake and microbial recycling of P and Ca could explain why neither the elemental stoichiometries of freshwater fish nor human feces were preserved in nutrient pools, whereas sediment stoichiometries were. It is more plausible, however, that human-derived inputs represented a recent fraction of ADE's chemical makeup, an alternative hypothesis that is supported by several independent lines of evidence.

To make this point abundantly clear, we now state in the conclusion that our findings do not preclude some human effects on the local environment, but soil properties are most parsimoniously explained through natural processes that precede the timing of plant domestication and land management in the region. Notably, this conclusion is supported by the reviewer's own suggested references listed below.

(ii) Reviewer 1 also argued that a flaw in our study is that we failed to consider the *"accurate chronology of human occupation"*.

To address the issue of chronology of occupation, the reviewer urged us to consider 20 references which included peer-reviewed and non-peer-reviewed publications. We carefully read and considered each of the suggested references to prepare a detailed list of outcomes from each publication, which strengthened our analysis and offered further support for the chronology proposed in our original submission (see color-coded list below).

Among the reviewer's 20 references, we found two major peer-reviewed synthesis papers that we had already cited as the basis for our chronology of human occupation (Maezumi et al 2018 and Bush et al 2016). Those studies were entirely consistent with the chronology described in our original submission. Here is a summary of main findings that corroborate our interpretation.

Maezumi et al. The legacy of 4,500 years of polyculture agroforestry in the eastern Amazon. Nature Plants (2018) The authors of this synthesis concluded that *"in the eastern Amazon, the subsistence basis for the development of complex societies began ~4,500 years ago with the adoption of polyculture agroforestry, combining the cultivation of multiple annual crops with the progressive enrichment of edible forest species"*. This is precisely the same timeline used in our original manuscript. Here is a quote from our previously submitted text in which we cite the referred study: *"we found that large exogenous inputs of both microcharcoal and mineral elements, preceded the emergence of complex societies and cultivation of edible species in the region"⁵⁵*; that is, before 4,500 years BC (by ~3,000 years; Fig 3). In our original conclusion we also summarized the phases of occupation identified by Maezumi et al in our conclusion. Here is a quote: *"The most recent evidence of ancient human occupation across the Amazon basin delineates three main phases of soil management: a pre-cultivation period (>6000 years ago); an early-cultivation period (6000–2500 years ago); and a late-cultivation period (2500–500 years ago)"*; that is, precisely the same phases described by Maezumi et al. Moreover, the regional archaeological and palynological synthesis shows no previous record of Maize cultivation near our site (Fig 1 white circles in Maezumi et al). The nearest site in which Maize pollen occurs is located ~600 linear kilometers to the East where the earliest Maize pollen is found ~4,000 years BP (Maezumi et al). As a result, we can rule out early Holocene Maize cultivation at our site where we found that ~19% of the microcharcoal pool in deep ADE profiles is C4-derived (Fig 4) and ~7,500 years old (Fig 3). In summary, our interpretation is consistent with the accepted chronology of land management in the region, which allows us to rule out cultivation as the local source of ADE carbon and mineral elements during the early Holocene.

Bush et al A 6900-year history of landscape modification by humans in lowland Amazonia. QuatSciRev (2016) The authors of this review concluded that *"High-resolution fossil pollen, diatom, and*

charcoal, record from the Peruvian Amazon that spans 6900 years” in a region that is thought to be the cradle of Amazonian societies in the Peruvian Andes (thousands of kilometers away from our site). The date 6900 BP is earlier than the chronology proposed for Eastern Amazon by Maezumi et al, but it is still not as early as our charcoal record (>7500 BP). Notably, we did cite this study in our original submission and explained that even in the Peruvian Amazon Bush et al found that “Evidence of intensified agricultural production [including Maize cultivation] between c. 3380 BP and 700 BP”, which is entirely consistent with the widely accepted chronology of land cultivation. Unfortunately, Reviewer 1 overlooked the second paragraph of our introduction where we cite Bush et al and explain that: “Some of the earliest evidence of permanent settlements in Amazonia come from riverine areas near the Peruvian Andes where, due to the limited agricultural potential of indigenous soils, food cultivation occurred on constructed terraces⁷. Those settlements left marked records of increasing fire disturbance, deforestation, and erosion, as well as non-forest-derived domesticated plants (e.g. fossil maize pollen) in soils that have been managed for much of the past 6900 years^{16,17}. Notably, ADEs were never found near those early Andean settlements¹⁸. Indeed, the vast majority of ADE sites were found thousands of kilometres away, in the central and eastern Brazilian Amazon¹⁹, where evidence of human occupation is much more recent”.

Other significant comments

(iii) *“The authors cite the eastern Amazon, Santarém region as their reference for human chronologies, using the apex of pre-Columbian activity (starting 4,500 and peaking at 2,500 BP). As a correction to the data used in this study, in the eastern Amazon, the earliest evidence of human activity is ~11,000 cal BP from Pedra Pintada Cave, Monte Alegre, Pará. There is extensive archaeological evidence led by Eduardo Neves (main advisor of MS and PhD theses listed above) that is closer to the study region that would be more relevant than the eastern Amazon case study. There is a high density of archaeological sites near the EMBRAPA study area (>5 archaeological sites less than 10 km away and more than 20 sites within a 30 km radius)². These records are essential to the interpretation of the ADE histories in the area as the earliest evidence of local human activity is >9,000 years ago 2–6 and in the broader region >11,000 years ago^{1,7,8}. If the accurate chronology of human occupation in the region is considered, it is not possible to exclude humans as the dominant driver of ADE formation.”*

The excavations mentioned here showed a diversified lithic industry, including bifaces, projectile points and dates back to the early Holocene. Those surveys demonstrated that the best places to find evidence of hunter-gatherers in the region were the sand-rich areas where outcrops of silicified sandstone occur near the streams belonging to the Rio Negro basin. As the authors of the theses explain, those were non-ceramicist hunter-gatherer communities, not land managers or agricultural communities. It is well known that human activity in the Amazon dates back to the early Holocene. However, the invention of agriculture and soil management is much more recent. We now clarify this point in our introduction and make a clear distinguish between the chronologies of occupation and the timing and types of land management (see also new references and discussion in response to Reviewer 2). For example, the archeological artefacts that are related to land management dated by Helena Pinto Lima, Eduardo Neves & James B. Petersen (2006) near our site in Hatahara (13 in the figure below) range between 960±40 BP at 155 cm depth and 1300±40 BP at 192 cm. In site number 12 (Lago Grande) at the opposite side of Caldeirão, dating of ceramic artefacts is 1940±60 BP at 100 cm depth. Those ceramics were apparently related to the Guarita phase of the Polychrome tradition which appeared in the lower Solimões region around 1,000 AD, around 800 AD in its tributaries near the city of Coari, and even earlier during the fifth century AD in the Tefé area. Source: Macedo et al. Amazonian dark earths in the fertile floodplains of the Amazon River, Brazil: an example of non-intentional formation of anthropic soils. *Bol. Mus. Para. Emílio Goeldi. Cienc. Hum.*, Belém, v. 14, n. 1, p. 207-227, 2020.

Figura 1 – Localização dos sítios arqueológicos citados no texto (02 – Açutuba; 03 – Cachoeira; 09 – Osvaldo 09; 12 – Lago Grande; 13 – Hatahara).

Source: Helena Pinto Lima, Eduardo Neves & James B. Petersen La fase Açutuba: um novo complexo cerâmico na Amazônia Central. In: Cristóbal Gnecco y Alejandro Haber. Arqueologia Suramericana. Departamento de Antropología, Universidad del Cauca & Facultad de Humanidades, Universidad Nacional de Catamarca. World Archaeological Congress, v. 2(1):26-52, 2006.

(iv) “Another line of evidence used to support riverine inputs is from the isotopic enrichment of ADE carbon profiles that coincides with pyrogenic carbon inputs. Based on the revised archaeological chronology discussed above, this will co-occur with human occupation in the area. While the enrichment of $\delta^{13}C$ values could indeed be linked to the expansion of the C4-dominated “campinaranas”, the presence of humans at the study site, necessitates the consideration that crop cultivation may have influenced the $\delta^{13}C$ values both indirectly (through the removal of trees for crop cultivation which increased C4 grasses) and directly (through planting C4 crops such as maize). Previous research has shown that pre-Columbian populations used open landscapes for crop cultivation because it was less labour intensive than clearing forested landscapes¹⁸. Additionally, maize cultivation is often preceded by a peak in sedimentary charcoal¹⁶. There are numerous examples of concurrent maize cultivation and fire management in the region dating to the genesis of the ADE soils at the EMBRAPA study site^{15,16,19}”

As explained above, the regional and local archaeological and palynological data show no previous record of Maize cultivation at or near our site (e.g. Fig 1 white circles in Maezumi et al). The nearest site in which Maize pollen occurs is located ~600 kilometers to the East and dates ~4,000 years. This finding also addresses a question posed by Reviewers 2 and 3, both of whom wondered if we considered the presence of Maize cultivation. The answer is yes, we did, and we interpreted C4-derived pools as exogenous because there is no evidence of grass or Maize phytoliths at the site. For example, Macedo et al performed phytolitic analyses in ADE samples taken each 5 cm depth at our study site. They found that the overall number of phytoliths of *Arecaceae* and *Cyperaceae* species is higher in ADE than non-ADE horizons, but found no sign of domesticated species. They concluded that “the absence of phytoliths [of agricultural species] indicates that the ADE genesis was not related to agricultural practices” (Source: Macedo 2014. Pedogênese e indicadores pedoarqueológicos em terra preta de índio no município de Iranduba – AM. In Portuguese). The same conclusion was reached at the Hatahara site near our site, see

map above (Source: Bozarth et al. Phytoliths and *Terra Preta*: The Hatahara Site Example. In: Amazonian Dark Earths: Wim Sombroek's Vision pp 85-98). Therefore, we can rule out early Holocene Maize cultivation at our site where we found that ~19% of the microcharcoal pool in deep ADE profiles is C4-derived (Fig 4) and ~7,500 years old (Fig 3). One alternative explanation proposed by Reviewer 2 is that our record can be interpreted as the earliest signal of Maize cultivation in the study region. We explore this possibility in the revised submission (see response to Reviewer 2) while emphasizing that exogenous inputs of carbon is a more parsimonious explanation for two reasons. First, due to the multiple signals of alluvial deposition found in the elemental profile of the ADE site in association with differences in particle size distribution, microcharcoal abundance, and isotopic signatures. Second, the absence of any Maize pollen in the region and the absence of grass or Maize phytoliths at our site (as discussed above).

(v) Among the remaining 18 references proposed by this reviewer 8 are non-peer-reviewed dissertations or textbooks, for which we were unable to find corresponding peer-reviewed publications. The 10 peer-reviewed sources offer direct support for the chronology of land use and a few bring new information about human occupation and fire or carbon dynamics before plant domestication, which we included in our revised submission.

Summary of publications suggested by Reviewer 1. Red – non-peer-reviewed theses (not cited); Green – non-peer-reviewed books (not cited); Purple – peer-reviewed sources (most of which are cited in the main text).

1. Roosevelt, A. C. et al. Paleindian cave dwellers in the Amazon: the peopling of the Americas. *Science* (80-.). 272, 373–384 (1996).

An interesting but off topic paper published in 1996 about “early foragers” “cave dwellers” at a site located 700 linear kilometers away during the Pleistocene; that is, before the invention of agriculture and soil management.

2. da Silva Costa, F. W. *Arqueologia das campinaranas do baixo Rio negro: Em busca dos pré-ceramistas nos areais da Amazônia central.* (Universidade de São Paulo, 2009).

A PhD thesis defended in 2009 which apparently did not result in peer-reviewed publications according to a google scholar search and a search for “trabalhos decorrentes” at the USP university repository (<https://www.teses.usp.br/teses/disponiveis/71/71131/tde-29072009-145147/pt-br.php>). This thesis was advised by a researcher whose more recent work we cited in our original submission.

3. Zuse, S. *Variabilidade cerâmica e diversidade cultural no alto rio Madeira, Rondônia.* (University of São Paulo, 2014).

A PhD thesis defended in 2014 which apparently did not result in peer-reviewed journals according to a google scholar search and a search for “trabalhos decorrentes” at the USP university repository <https://www.teses.usp.br/teses/disponiveis/71/71131/tde-25092014-124246/pt-br.php>

4. Caldarelli, S. B. & Kipnis, R. *A ocupação pré-colonial da Bacia do Rio Madeira: novos dados e problemáticas associadas.* *Especiaria - Cad Ciências Humanas* 17, 229–89 (2017).

Publication in obscure journal with questionable 14C dates of bulk ceramic fragments (the actual source of carbon is unclear) collected ~1050 kilometers away (SW), showing numerous dates that are consistent with our interpretation and 3 dates that are older than expected. See Table 5 n= 5; Date range 1160-3910 years BP. Table 7; n= 38 Date range 110-6050 years BP and n=3 dates >7500 BP. file:///Users/Silva/Downloads/1767-Texto%20do%20artigo-7530-2-10-20180207.pdf

5. Mongeló, G. Z. *O Formativo e os modos de produção: ocupações pré-ceramistas no alto Rio Madeira, RO.* (University of São Paulo, 2015).

A MS thesis defended at USP in 2015 which apparently did not result in a peer-reviewed publication, advised by a researcher whose more recent work we cited in our original submission <https://teses.usp.br/teses/disponiveis/71/71131/tde-17092015-110653/en.php>

6. Lima, M. D. N. *O rio Unini na Arqueologia do baixo rio Negro, Amazonas. Dissertação.* (Universidade de São Paulo).

Another MS thesis (this one at a different institution – FAPESP) without apparent derived peer-reviewed publication, advised by a researcher whose recent work we cited <https://bv.fapesp.br/pt/dissertacoes-teses/93318/o-rio-unini-na-arqueologia-do-baixo-rio-negro-amazonas>

7. Capriles, et al. *Persistent Early to Middle Holocene tropical foraging in southwestern Amazonia.* *Sci. Adv.* 5, eaav5449 (2019).

An interesting recent study that is now cited in our revised manuscript. Notably, the authors state precisely the same timeline that we did for the emergence of complex societies, here is one quote: “The Amazon witnessed the emergence of complex societies after 2500 years ago that altered tropical landscapes through intensive agriculture and managed aquatic systems”.

8. Lombardo, U. et al. *Holocene land cover change in south-western Amazonia inferred from paleoflood archives.* *Glob. Planet. Change* 174, 105–114 (2019).

This is a fascinating recent study that is now cited in our revised manuscript. The authors provide direct support for the timing and magnitude of paleo floods which we propose as a possible mechanism for ADE formation, although the authors did not consider or mention ADEs in their study.

9. Amazonian Dark Earths Dordrecht: (Kluwer Academics., 2003).

In our manuscript we stated that “*Many scientific articles and books have been written about the site, all of which characterized the local ADE as a typical Anthrosol*”, this is one of those books.

10. Woods, W. I. et al. Amazonian dark earths: Wim Sombroek’s vision. (Springer, 2009).

Same as above. We stated that “*Many scientific articles and books have been written about the site, all of which characterized the local ADE as a typical Anthrosol*”, this is another of those books.

11. Maezumi, S. Y. et al. The legacy of 4,500 years of polyculture agroforestry in the eastern Amazon. Nat. Plants (2018)

We already cited this paper in our original submission. It brings a regional synthesis of archaeological and paleo vegetation data upon which we based our data interpretation

12. de Souza, J. G. et al. Climate change and cultural resilience in late pre-Columbian Amazonia. Nat. Ecol. Evol. 3, 1007–1017 (2019).

Interesting recent study that shows no significant departures from the chronology of occupation that we used and is now cited in the discussion to explain uncertainties of population density estimates

13. Hilbert, L. et al. Evidence for mid-Holocene rice domestication in the Americas. Nat. Ecol. Evol. 1, 1693–1698 (2017).

Another fascinating paper that offers further support for the chronology of occupation that we used (see point 2 above)

14. Watling, J. et al. Direct archaeological evidence for Southwestern Amazonia as an early plant domestication and food production centre. PLoS One 13, e0199868 (2018).

Questionable 14C dates interpretation for charcoal collected ~1050 kilometers away (SW), showing only 1 early Holocene carbon date described as “charcoal plotted in profile” and 4 actual soil carbon dates that are consistent with mid Holocene cultivation (Table 1 <https://journals.plos.org/plosone/article?id=10.1371/journal.pone.0199868>)

15. Bruggner, S. O. et al. Long-term man–environment interactions in the Bolivian Amazon: 8000 years of vegetation dynamics. Quat. Sci. Rev. 132, 114–128 (2016)

This is another review that brings a series of chronologies which offer further support to our interpretation. The authors found that “*Pollen and charcoal suggest agriculture and human use of fire since 6500 cal BP*” (now cited in the revised text)

16. Bush, M. B. et al. A 6900-year history of landscape modification by humans in lowland Amazonia. Quat. Sci. Rev. 141, 52–64 (2016).

We already cited this paper in our original submission

17. Lombardo, U., Denier, S., May, J.-H., Rodrigues, L. & Veit, H. Human–environment interactions in pre-Columbian Amazonia: The case of the Llanos de Moxos, Bolivia. Quat. Int. 312, 109–119 (2013).

This is another study conducted in Peru thousands of kilometers away from our site, which brings no significant departure in terms of timing of occupation and land use in relation to several more recent studies in the same region

18. Carson, J. F. et al. Pre-Columbian land use in the ring-ditch region of the Bolivian Amazon. The Holocene 25, 1285–1300 (2015).

This another good study which we wish to cite in our revision. Here too the authors found “*First occupation around the lake was radiocarbon dated to ~2500 calibrated years before present (BP)*”, thus not altering our interpretation.

19. Kistler, L. et al. Multiproxy evidence highlights a complex evolutionary legacy of maize in South America. Science (80-.). 362, 1309 LP – 1313 (2018).

This is a fascinating recent study now cited in the revised submission. Here too the chronology of occupation including the earliest evidence of maize cultivation in the Brazilian Amazon is entirely consistent with Maezumi et al 2018 (~4,000 years BP; see point 2 above).

20. Arienzo, M. M., Maezumi, S. Y., Chellman, N. J. & Iriarte, J. Pre-Columbian Fire Management Linked to Refractory Black Carbon Emissions in the Amazon. Fire vol. 2 (2019).

This is a fascinating recent study that is consistent all other papers by Maezumi et al.

Overview: The reviewer stated that *“This is an excellent piece of work, very detailed and clearly presented to the point where I could not identify a typographical or grammatical error to correct (which is very unusual) ... the authors have made a solid case that natural processes have contributed to site formation”*, however, the reviewer also suggested *“some reworking to tone down the over-writing of humans as important in the genesis of ADEs. I am convinced by the arguments that there is an exogenous input to the nutrients at the site – that is clear from the data presented. The arguments about a smaller human imprint are weaker”*.

We are thankful for this reviewer’s positive assessment of our manuscript. We closely followed the reviewer’s many insightful suggestions, which improved our narrative and interpretation of results. Here we list the most important changes we made in response to this reviewer’s comments. A full list of revisions can be found in the enclosed annotated PDF.

(i) *“There is a single paper from 2003 (line 245) referred to as evidence that populations were not sufficient to generate the nutrient inputs calculated. There has been a lot of work more recently that has demonstrated considerably more evidence for widespread occupation of the Amazon basin, so I am unconvinced the population densities weren’t higher than proposed in that 2003 paper”*

We appreciate the reviewer’s concern and we have improved this section to clarify our population density estimates as well as to consider potential sources of uncertainty. For example, we included the caveat *“it is important to note that population density estimates have large margins of error and should not be used to generalize patterns across the region”*. We also added new references to explain that *“The Amazon basin has a complex history of occupation and land use. Prior to European contact, indigenous peoples relied on >80 plant species whose distribution was associated with settlements that covered ~0.1% of the region²¹. A recent analysis of cultural transitions suggests that specialized land-use systems faced major reorganization during Holocene climate fluctuations, whereas others that relied on polyculture agroforestry were more resilient and remained unaffected⁶⁹. Our detailed dataset of input amounts and sources is site-specific and does capture regional transitions in land use. However, the clear picture emerging from our data is that indigenous populations discovered how to identify and use ADEs but were not responsible for their genesis, a hypothesis that can be tested in future regional studies”*

To minimize the uncertainty associated with different land use strategies, we used the most conservative estimates of human input, based on the richest possible source of P and Ca typically found in ubiquitous waste materials. Additionally, we assumed that no nutrient losses occurred after the deposition of waste materials. If we were to account for other nutrient sources or losses, either the total number of people per area or the total time of continuous occupation would have had to be orders of magnitude larger than previously reported in the anthropological literature. We explain our approach and provided several supporting references in the methods, as in, for example, *“...fish waste is the richest of those sources, ranging from 2-4% and 3-6% of the whole fish on a mass basis³². Thus, it yields the most conservative population densities”* and, in the discussion, *“To minimize uncertainties related to type and rate of biomass deposition we used the most conservative estimate of inputs available. Specifically, we considered three ubiquitous anthropic sources of Ca and P (i.e. faeces, plant, and fish biomass). Of these sources the latter is by far the richest source of Ca and P on a mass basis, and thus it yields the most parsimonious estimate of waste needed to build soil fertility”*.

To convert biomass input into approximate number of people we used population densities estimated for riverine areas where ADEs are typically found. In addition to citation 22, mentioned by this reviewer (i.e. the 2003 paper by Denevan *“The native population of Amazonia in 1492 reconsidered”*) we included several more recent papers that support our choice of population density end members. Here is a fragment of the revised text and new references *“... a synthesis of anthropological data in those areas [riverine sites where ADEs are common] suggests population densities ranging from 20 to 60 people per hectare*

prior to European contact²⁰. A recent re-evaluation of anthropological methods used to obtain those estimates corroborates those same densities and suggests that at least five million people inhabited the Amazon region in 1492⁶⁴. Both total population and settlement densities are consistent with the earliest observations of native peoples in South America⁶⁵ and fall within the range of dense pre-Columbian forested landscapes elsewhere in the Americas. For example, in 1519 the Aztec empire was comprised of several million people, living in city-capitals of median density ~50 people per hectare⁶⁶”.

Finally, using 20 to 60 people per hectare as end members, and using a series of detailed observations of indigenous diet we explain that “*each adult consumed, on average, a maximum of 219 grams⁶⁷ and a minimum of 90 grams of fresh fish per day⁶⁸”* to calculate that the formation of a small ADE patch of 1 hectare would have required ~1000-8000 years of continued occupation. By the same logic, if human faeces were the sole source of nutrients, “*the time necessary for ADE formation would be ~1500 to 18000 years, assuming that each adult eliminates on average 128 g of fresh faeces containing ~75% water, and 7.5 to 30 mg kg⁻¹ day⁻¹ of P and Ca in the remaining dry mass³³”*. Therefore, even if the highest ever recorded population densities are considered for the 12-hectare patch that we sampled, ADE formation would have required several millennia of continuous nutrient input.

We conclude by explaining that “*it is possible that dense populations occupying riverine areas uninterrupted for thousands of years were capable of recycling nearly all of their waste into ADEs, where little to no loss through leaching or burning occurred. It is also possible that inherent differences in rates of plant uptake and microbial recycling of P and Ca could explain why neither the elemental stoichiometries of freshwater fish nor human faeces were preserved in exchangeable or total nutrient pools. It is more plausible, however, that human-derived inputs represented a minor fraction of ADE’s chemical makeup”*

(ii) “*The authors claim there is no evidence for occupation early enough to explain the radiocarbon dates at depth in the ADE. Lombardo et al. (2020) Early Holocene crop cultivation and landscape modification in Amazonia. Nature, recently suggest dates around 10,000 years ago for cultivation. This is admittedly in the western Amazon, but I think absence of evidence does not equate to evidence of absence in this case. If we accept 10,000 years then this is within the range of plausible nutrient additions calculated by the authors (line 228)”*

We thank the reviewer for suggesting the Lombardo et al 2020, which we had not seen before our original submission. We cited the study in the introduction of our revised manuscript as an improvement to our timeline of occupation. It is important to note that that study took place in the Llanos de Moxos region in a savanna-dominated landscape located in Bolivia (thousands of linear kilometers SW from our study site). The authors of that study found “*evidence of manioc-cassava, yucca 10,350 years ago, and squash 10,250 years ago”*. However, consistent with our interpretation of exogenous C4-derived carbon inputs (explained in response to Reviewer 1), the study also found that “*Early maize appears 6,850 years ago”*. Therefore, our C4-derived microcharcoal is at least ~1000 years older than the earliest ever recorded dates of Maize cultivation and, as such, we can rule out that type of cultivation as the genesis of our ADE site.

We also added language to differentiate central Amazonia from the Andean Amazon, which is comprised of a different set of ecosystems and cultural traditions than those in our rain forest region. We now clarify that “*many Amazons”* existed before European contact from the perspective of social and ecological systems. That is to say that we are not denying that humans arrived in the Amazon region 11,000 - 10,000 years ago, but there is no evidence that they settled and managed landscapes in our study region before the mid Holocene. We added new peer-reviewed references (suggested by Reviewer 1) to describe some of different practices that differentiate the chronologies of occupation and land use. There is no doubt that human presence in the region began during the early Holocene, but cultivation and land management in our region is much more recent. For example, the oldest ceramic artifacts at our site and in neighboring

archaeological sites (see map above) date from 960±40 BP at 155 cm depth and 1300±40 BP at 192 cm depth (Source: Lima et al. The Açutuba phase: a new ceramic complex in Central Amazon (in Portuguese). Ceramics from Açutuba phase have been excavated (100 - 155 cm soil depth) from nearby archaeological sites and date between 960±40 BP and 1940±60 BP (Macedo et al. Amazonian dark earths in the fertile floodplains of the Amazon River, Brazil: an example of non-intentional formation of anthropic soils in the Central Amazon region. Bol. Mus. Para. Emílio Goeldi. Cienc. Hum., Belém, v. 14, n. 1, p. 207-227, 2020). The earliest evidence of human activity in eastern Amazonia was recorded in Pedra Pintada Cave ~11,000 years BP in. However, those were small hunter-gatherer communities compare to large densely populated settlements in central Amazon ~1,000 BP, which are thought to have had ~2,000 people supported primarily through the exploitation of aquatic resources and C₃ plants. For example, archaeological evidence suggests that society was stratified at the peak of the Marajoara phase (~700–1100 BP). The elite lived on large mounds, controlling access to prestige ceramics and water-management systems. Mounds in the flooded savannas reached ~3 ha in area and 7 m height. Population estimates are up to 2,000 for a mound group. Unlike the Arauquinoid, there is no evidence that the subsistence of the Marajoara depended on cultivated plants. Macro-botanical remains of maize are absent, and human bone isotopic values indicate a diet based on non-domesticated C₃ plants and aquatic resources” Souza et al. Climate change and cultural resilience in late pre-Columbian Amazonia. Nature Ecology & Evolution, v. 3, p|| 1007-1017, 2019. (now cited in the text)

We have synthesized and clarified the points above by differentiating the chronologies of occupation from the chronologies of land management, adding several new references, as in, for example: *“Recent findings suggest that the chronology of plant domestication dates back to >10000 years ago in western Amazonia¹⁸, with the emergence of complex societies that relied of intensive agriculture and management of food systems occurring <4000 years ago¹⁹. Some of the earliest evidence of permanent settlements in Amazonia come from riverine areas near the Peruvian Andes where, due to the limited agricultural potential of indigenous soils, food cultivation occurred on constructed terraces¹⁰. Those settlements left records of deforestation and erosion as well as domesticated plants in soils that have been managed for much of the past 6900 years^{6,7}, although for most of the Amazon basin the earliest evidence of intensive cultivation inferred from pollen data falls between 3380 and 700 years ago⁵. Notably, ADEs uncommon near early settlements of the western Amazon¹⁸. The vast majority of ADE sites are found thousands of kilometres away in the central and eastern Brazilian Amazon¹⁹, spatially associated with riverine bluffs¹⁷, where evidence of management is much more recent (2500-500 years ago)²”*

Finally, in response to this comment as well as other reviewers, we refined our main conclusions to better represent the importance of human inputs. For example, the revised abstract now reads *“Multiple lines of evidence point to the alternative hypothesis that ancient populations harnessed dynamic processes of landscape formation, which led to the unique properties of ADEs, but were not responsible for their genesis. This implies that indigenous populations used their knowledge of the landscape to identify and favourably settle areas of exceptionally high fertility prior to the emergence of intensive plant cultivation and soil management in central Amazonia”*. Additionally, we now state in the conclusion that *“Our findings do not preclude some human effects on the local environment. Indeed, the wisdom of native populations, manifested for example in the application of waste materials to the soil²⁴, may well have further enriched ADE profiles or, at least, countered their otherwise-inevitable degradation”*

(iii) I think the evidence of C₄ biomass in TOC could equally be taken as evidence of human modification of the local landscape, it is not diagnostic either way.

Typically, we would agree with this assessment. However, the timing of the C₄ input and its consistency with the paleoflood archives (discussed above) tips the scale towards exogenous input. We expanded the rationale for this interpretation as follows *“Land management for cultivation could have influenced the $\delta^{13}\text{C}$ values both indirectly (through the removal of trees) and directly (through C₄ crops such as maize).*

However, the nearest area where Maize pollen is found (~600 linear kilometres to the East of our site) shows the earliest evidence of cultivation ~4,000 years ago⁵. Additionally, in response to this comment and those of other reviewers we added several new references that provide additional support for the interpretation of C₄-derived carbon at this site as exogenous inputs. For example, we now explain that “previous studies yielded no evidence of Maize phytoliths at our site or in other typical ADEs in the region ... a detailed analysis of Pre-tic horizons found that Maize phytoliths are absent at our site³⁷ and did not appear in the regional record until the late Holocene³⁸⁻⁴⁰. Therefore, we can rule out early Holocene crop cultivation as the source of C₄-derived carbon input at the onset of ADE genesis” [see response to reviewer 1 for a map and additional references]. Therefore, both in situ and regional evident support that riverine inputs were the source of C₄ carbon [see paleo flood archives for details].

(iv) “Its had to tell from the way the data is presented, but it appears that charcoal concentration increases fairly dramatically up the profile – this is consistent with an increasing intensity of use as the surface is built up, with the trend in radiocarbon dates suggest there has been aggradation. Given the resistance of charcoal to degradation I would expect similar concentrations down the profile if this was due to natural processes? The same would not be true for TOC which degrades over time”

We chose to present charcoal concentration as a fraction of total organic carbon to be consistent with previous studies of this kind. The reviewer is correct about the resistance of charcoal relative to total organic carbon. In our study, this is illustrated by the mean residence time (age offset) of bulk and charcoal carbon, which increases with depth along the 100 cm profiles regardless of soil type (Fig 5). The reviewer is also correct that the trend in radiocarbon dates suggests aggradation, however, our data cannot be used to infer the amount of charcoal contributed to the system at different points in time because of physical and chemical degradation. Although more recalcitrant than bulk organic matter, pyrogenic-C is degradable and tends to be transformed from macro to micro particles (Lutfalla, 2017 Pyrogenic carbon lacks long-term persistence in temperate arable soils. *Frontiers in Earth Science*, v. 5, article 96, 2017 – now cited in the discussion). This could explain why the largest relative contribution and total amount of pyrogenic microcharcoal at our site is found at intermediate depths. Furthermore, relatively recent inputs from human activity may have contributed additional charcoal into the soil matrix, and thus the 14C dates that we report probably represent the most recent of a range of possible ages of charcoal input. This means that the earliest input event that we found, which is from a different source (i.e. C₄-derived deep soil) than recent inputs associated with human activity (i.e. C₃-derived top layers), are in all likelihood significantly older than 7500 years. We now describe this process in the discussion and methods, explaining why it offers further support for the interpretation of exogenous inputs prior to human activity.

Other points

“Right until the end, I thought ‘exogenous inputs’ meant nutrients from seasonal flooding, which does make sense to me, but then I learn in the methods that the surface is 40 m above the river level. I suspect the annual flood pulse is not that high (maybe 10m?) and rates of uplift/incision are fairly low, so I am left unsure how the excess nutrients are thought to have accumulated. This needs to be clarified – is it excess nutrients from some time in the early Quaternary when the site was an active seasonally inundated feature, in which case I am unsure why extra Ca and P would not have had time to leach away a long time before the Holocene (since the authors claim Ca leaching can be rapid at line 238)”

As explained above, we are referring to exogenous inputs prior to human activity. We clarified this point throughout the text. For example, we now explain in the discussion that “the alluvial plain of the Solimões-Amazon River, optically stimulated luminescence and geomorphological data show large fluvial bars of weathered sediment deposited at the Pleistocene-Holocene boundary⁵³” (but the deposition of carbon and nutrients measured in up to 100 cm depth is more recent, as described above, beginning during the early- to mid-Holocene transition. Further, we now explain in the methods that “The site is located just north of the Solimões River near its confluence with the Negro River, within the sedimentary

basin of the Amazon River. Satellite and field measurements show that the terrace is located ~40 m above the modern sea level, at a maximum of ~27 m above the modern river level, or ~10 m above the adjacent flood plain” (we also included a new figure - 7 – to illustrate the gradient and to address reviewer 3’s comments, see below). Therefore, excess nutrients from some time in the early- to mid-Holocene was likely given what we know about active river dynamics from other inundated feature. We also explain this in the revised text, for example, “the trace elements found at the same depths in which we see enriched P, Ca, and microcharcoal suggest major riverine inputs (e.g., Ni, Rb, Ti, Zn; **Table S1**). This implies that two critical ingredients needed for ADE formation (i.e. microcharcoal, which confers its colour and cation exchange capacity; and mineral fertility, including macronutrients and trace elements that are in extremely low supply in adjacent soils) were deposited at the site before the invention of agriculture ... deposition at our site more closely match those found in sedimentary deposits that can be traced to early-Holocene open vegetation fires upstream⁴¹ ... soil biogenic silica⁴⁶ and records of monsoon intensity (i.e. speleothem oxygen and strontium isotopes)⁴⁷, indicate a climate-driven shift in river dynamics following a regional increase in precipitation after a persistent dry period (~8000 to ~4000 years ago)⁴⁵”

(ii) “Round line 238 – this suggests that Ca should be rapidly removed, so will not accumulate and cites evidence for observed Ca leaching from ADEs. This I think could equally be interpreted as close cycling of Ca during a previous period of intense management, with little loss, changing following abandonment to losses for Ca under conditions of no management. In term of P, I assume the reference is to loss as fine particulates, but again, depends on how it was burnt, ie management, and losses may not have been high from this mechanism”

We agree with this suggestion and added language and references to reflect this possibility in the discussion.

Response to Reviewer 3

Overview: The reviewer stated that ADEs “are associated with archeological deposits and have attracted a lot of work and comment but have seldom been well characterized by soil stratigraphers and soil geochemists. Nor have they been well studied by soil geomorphologists”. The reviewer also stated that our “manuscript argues that rather than native people control on ADE development, the ADEs formed by natural fluvial/pedogenic processes and were exploited later by people”, which we are also in agreement with. The reviewer’s main criticism can be organized around two themes (i) interpretation of data and (ii) sources of uncertainty and alternative explanations for the observed results.

We agree with this assessment. Those are indeed areas that needed improvement. We carefully addressed this reviewer’s suggestions in tandem with related comments provided by reviewer 2. Please see responses above, which complement the answers below.

(i) “the ms seems to draw conclusions not merited by the data and/or uses the wrong data to evaluate a point. For instance, although the ADE and comparison Ultisols are close together it is not immediately obvious whether they are derived from different alluvial deposits separated in time. The ms argues that indeed they are the same deposit and uses the similarity in clay mineralogy (not elaborated) to argue that point. In fact clay mineralogy is the most likely component of the soils to converge to similarity over time and so it would be much more useful to use sand mineralogy to look for similarity/difference. Much of the interpretation depends on 13C and 14C data. In the case of the former the difference between the ADEs and Ultisol values is too small to be reliable given the potential differences in water use efficiency of different plants and the slight shift toward more heavier C in the ADE is interpreted to mean that savanna grasses grew on the site in the past”

We argue that the ADE is comprised of parent material that is a mixture of the upland ultisols and alluvial

material of Holocene age. A complex geomorphic history for the area summarized in a recent review (Passos, M. S. *et al.* Pleistocene-Holocene sedimentary deposits of the Solimões-Amazonas fluvial system, Western Amazonia. *J. South Am. Earth Sci.* **98**, 102455 2020) indicates that the site occurs at the lowest elevations of the Intermediate Terrace and was likely inundated by shallow waters during periods of higher regional river base level. We moved clay evidence to be subordinate to the isotopic evidence, which clearly supports the ADE as a mixture of Ultisol and alluvium. Additionally, we included several other references and revised our interpretation of results to include the reviewer's suggestion. For example, this clay mineralogy section of the revised manuscript now reads:

“Clay mineralogy characterization using X-ray diffraction showed no differences in the type of dominant clays found in ADE and Ultisol profiles (Fig S2), but that can be explained by two factors. First, across the alluvial plain of the Solimões-Amazon River, optically stimulated luminescence and geomorphological data show large fluvial bars of weathered sediment deposited at the Pleistocene-Holocene boundary⁶⁰ (i.e. long before the deposition of carbon and nutrients described above). Second, no neo-formation of minerals is expected for the highly-weathered assemblages (e.g. kaolinite and iron/aluminum oxides, such as goethite) that are abundant in river sediments of this section of the Amazon basin⁶¹ and dominate regional soils⁶². Thus, it is not surprising that clay mineralogy is the only measured variable to converge in ADE and Ultisol”

60. Sant'Anna, L. G., Soares, E. A. do A., Riccomini, C., Tatum, S. H. & Yee, M. Age of depositional and weathering events in Central Amazonia. *Quat. Sci. Rev.* **170**, 82–97 (2017).

61. Guyot, J. L. *et al.* Clay mineral composition of river sediments in the Amazon Basin. *CATENA* **71**, 340–356 (2007).

62. Quesada, C. A. *et al.* Soils of Amazonia with particular reference to the RAINFOR sites. *Biogeosciences* **8**, 1415–1440 (2011).

(ii) Seemingly neglecting the possibility of OM contribution from corn grown on the site. The mean residence time $\delta^{13}\text{C}$ values for charcoal in the ADEs and the Ultisols are similar, although charcoal is in greater abundance in the ADE. The explanation for the similarity and difference between the two soils and their underlying stratigraphy depends on the use of Sr and Nd isotopes to draw linkages between the ADE isotopic signatures and that of river sediment and then there is an unsatisfying linkage drawn with sites much further upstream. This explanation is key to the paper and is not at all well explained. Without it the rest of the ms is enticing but empty. We are asked to take this site as iconic for many other ADEs but the linkage is not made – I suspect it is there and lies in the location of the ADEs along bluffs at the edges of rivers but even the context of the site under consideration to the existing nearby river is not presented (how far above flood level is the site? What is the down cutting rate of the river? How close to the river flood level could the river have been at the time of hypothesized inundation? Or is it an ongoing process? Why focus on superposition of elements via floodwaters as opposed to imposition of a different fluvial strata underlying the ADE?).

This is again a valid point that we have now more thoroughly addressed. As explained above we added a new paragraph and several references explaining that land management for cultivation could have influenced the $\delta^{13}\text{C}$ values both indirectly (through the removal of trees) and directly (through C_4 crops such as maize). However, we also explain that *“the nearest area where Maize pollen is found (~600 linear kilometres to the East of our site) shows the earliest evidence of cultivation ~4,000 years ago⁵. Moreover, previous studies yielded no evidence of Maize phytoliths at our site or in other typical ADEs in the region. For example, a detailed analysis of Pretic horizons found that Maize phytoliths are absent at our site³⁷ and did not appear in the regional record until the late Holocene^{38–40}”* Therefore, we can rule out early Holocene crop cultivation as the source of C_4 -derived carbon input at the onset of ADE genesis.

We also added a new paragraph and supplementary figures which show that *“The best estimate of terrain above modern river water level at the levee heights are a maximum of 27 m above the modern average river water level (i.e. ~10 m above the adjacent flood plain; Fig 7). Although above flood limits today, the*

site is at an elevation known to have been inundated in the early Holocene and possibly into the middle Holocene⁵³”.

Finally, we included a number of new references (including some from our own previous work on river and groundwater isotopic signatures) to explain that we do not expect a significant contribution from the modern river water or water table because Sr and Nd are mainly found in the mineral phase. For example, we now explain that “The concentration of Sr dissolved in river waters in the Amazon is in the order of parts per billion, whereas the Sr concentration in suspended sediments is on the order of parts per million⁵², and Nd is often below detection limits in river water⁵⁴. Thus, variation in Sr and Nd values were in all likelihood caused by fluvial mineral deposits that led to terrace development, which we interpret as the local manifestation of regional paleo floods identified in several other depositional sites^{55–58}”

52. Santos, R. V. *et al.* Source area and seasonal ⁸⁷ Sr/ ⁸⁶ Sr variations in rivers of the Amazon basin. *Hydrol. Process.* **29**, 187–197 (2015).
53. Passos, M. S. *et al.* Pleistocene-Holocene sedimentary deposits of the Solimões-Amazonas fluvial system, Western Amazonia. *J. South Am. Earth Sci.* **98**, 102455 (2020).
54. Bayon, G. *et al.* Rare earth elements and neodymium isotopes in world river sediments revisited. *Geochim. Cosmochim. Acta* **170**, 17–38 (2015).
55. Silva, C. L., Morales, N., Crósta, A. P., Costa, S. S. & Jiménez-Rueda, J. R. Analysis of tectonic-controlled fluvial morphology and sedimentary processes of the western Amazon Basin: an approach using satellite images and digital elevation model. *An. Acad. Bras. Cienc.* **79**, 693–711 (2007).
56. Quintana-Cobo, I. *et al.* Dynamics of floodplain lakes in the Upper Amazon Basin during the late Holocene. *Comptes Rendus Geosci.* **350**, 55–64 (2018).
57. Hayakawa, E. H., Rossetti, D. F., Hayakawa, E. H. & Rossetti, D. F. Late quaternary dynamics in the Madeira river basin, southern Amazonia (Brazil), as revealed by paleomorphological analysis. *An. Acad. Bras. Cienc.* **87**, 29–49 (2015).
58. Gonçalves, E. S., Soares, E. A. A., Tatum, S. H., Yee, M. & Mittani, J. C. R. Pleistocene-Holocene sedimentation of Solimões-Amazon fluvial system between the tributaries Negro and Madeira, Central Amazon. *Brazilian J. Geol.* **46**, 167–180 (2016).

Below I have specific comments.

L. 57-87. One of the key points about most ADEs is that they are spatially associated with river channels and as Denevan suggests may be even more associated with “bluff” habitations. This is an essential part of the story being presented here, if I understand right. The argument being laid out is that the ADEs have somehow been augmented by natural processes likely associated with fluvial processes (more on this below). It seems like it would make sense to start developing the spatial association with rivers and bluffs up front in these paragraphs.

We agree and incorporated Denevan’s Bluff model as requested in the introduction and discussion (see below).

L. 78-84. There are aspects of this sentence that don’t match with the evidence laid out by Macedo et al (2010) that the soils in western Amazonia are less weathered and higher in nutrients than those in central and eastern Amazonia. There is a concern here with the sweeping nature of some of the assertions in this paragraph. The idea presented here is that the soils everywhere are nutrient poor even though the broad evidence is more complex. Denevan was obviously working in a data-poor world with regard to soil properties. I don’t have a problem with the essence of the paragraph, but the building blocks have been mis-represented to some extent.

We agree and revised accordingly.

L. 104. Having trees there that are typically ~12 m high suggests that they must be regrowing secondary

forest rather than relatively stable forest communities, which makes sense if it is near an area that humans have utilized previously but a bit more nuance to the statement about dense rainforest with 12 m tall trees would be useful here.

That is the minimum canopy height for the forested area. We now provide more nuance to the statement explaining that “All intensively sampled soils were under dense rainforest (>12 m canopy height), which had not been managed for at least 40 years — i.e. since the EMBRAPA research station was established”

L. 109-113 and Figs. 1 and 2. The use of the term “exchangeable” with respect to P and Ca is incorrect usage if the ions in question if a Mehlich-3 extractant was used. There is no measure for P that is considered exchangeable and for Ca exchangeable would be expected to be determined by one of several standard methods such as NH₄Ac or BaCl. And I don’t understand the presentation of Ca as cmolc/kg if cation exchange was not measured (at least there is no mention of it). If Ca was measured on the Mehlich-3 extract, then it too should be presented as mg/kg in analogy with P and both should be called “extractable” rather than exchangeable. Fig 2 calls them “available” which is fine although it should be mentioned somewhere that you are assuming the Mehlich-3 extract is a measure of plant available ions. Also the last column of Fig. 2 has a funny ratio that should be rectified – if it turns out that Ca was actually derived from a measure of exchangeable extraction and therefore presented correctly in the second column then the values presented here should be transformed from molar to mass units prior to developing the ratio (remembering that what is being presented in the second column is the molar amount of charge that Ca is providing rather than a mass unit). Regardless the extraction technique(s) should be explained in the methods description rather than just mentioned in the fig caption. Fig 1 caption calls out Bottom Left and then describes the central left figure (which has been used in a previous publication and is fine but should be noted as such).

We accepted these suggestions. We revised Fig 2 and Fig 1, changed exchangeable to extractable throughout the text, and described the extraction method.

L. 132. In several places the term “rare mineral elements” is used which is confusing. I don’t know whether to interpret that as another way of saying rare earth elements which has a specific geological meaning or whether it is being used to denote all elements that are typically found in low abundance in crustal rocks and soils. If the latter I would suggest using trace element as a term and if the former to use rare earth element.

We accepted this suggestion.

L 140-148. A couple of points and questions. Given the list of 16 elements can you say anything about those that might be biocycled and those not? In passing, Se is a potentially interesting tracer because it is not much utilized by plants but required by animals and hence its distribution could be indicative of fish parts or not. It would be good to actually note what the clay minerals were. And frankly it is not surprising that the clay minerals are not different given that the formation of those minerals is guided to a great extent by climate with their amount being determined by time (or inheritance from parent material). So It would be much more instructive to quantify the sand or silt mineralogy to see if there were differences that might be indicative of either different parent materials or different lengths of weathering and hence potentially greater depletion of weatherable minerals in the Ultisols relative to the ADEs where the argument if I understand it right is that there has been augmentation.

We argue that a holistic view of the chemical and physical characteristics of ADEs is needed to decipher its genesis. We agree with the reviewer that it is not surprising that the clay minerals are not different given that the formation of those minerals is guided to a great extent by climate with their amount being determined by time. We added a new paragraph and new references to support that interpretation (listed above). We also agree that sand and silt content can be particularly revealing in this case and we added

discussion to this effect. For example, “support for the fluvial influence on the ADE soils is provided by differences in particle size distribution. Specifically, the classification of ADE profiles as “sandy clay loam” and adjacent Ultisols as “very clayey”¹⁰”. As for differential depletion in parent material we explain that “a first approximation, radiogenic signatures indicate that ~24% of Sr mass in ADE profiles originate from either fishbone or river sediment, as both sources have the same Sr isotopic composition”⁶⁰”. Finally, we agree that by focusing on individual elements much could be learned about different input sources. The analysis of Se in particular has long been a subject of interest in our region (e.g. Selenium content of Brazil nuts from two geographic locations in Brazil Chang et al Chemosphere, 1995). However, the cycling mechanisms of individual trace elements is beyond the scope of our study.

L. 152-183. There is a lot of important material in this paragraph, but I cannot fully grasp it. I get that the black carbon dates older than the bulk carbon. This makes sense if we assume that black carbon is a more recalcitrant fraction of the total. But one needs to be careful in the interpretation since all ¹⁴C values in soil are going to be mean residence time values. I don’t think there is a problem with the interpretation that the black C is older than when it is thought that people were there but is it not possible that it just means people were there earlier than we thought. After all the age of the peopling of the Americas keeps getting pushed older. That should be considered as possibility here even if you want to argue that the weight of evidence for the ADE formation lies elsewhere.

We agree and revised this section to clarify our point. In addition to revisions to the original text, we added two new paragraphs that make the distinction between the chronologies of occupation (early Holocene) and land management (late Holocene) more explicit. Indeed, the age of the peopling of the Americas keeps getting pushed older and our interpretation is well aligned with the most recent evidence of occupation from across the region (see responses to reviewer 1). Those paragraphs also answer similar comments were made by reviewer 2 (described above). Regarding the specific issue of carbon dating, we added several references to explain that our ¹⁴C dates represent the most recent of a range of possible ages because “inputs from human activity as well as natural processes may have contributed additional charcoal into the soil matrix. Moreover, although more recalcitrant than bulk organic matter, charcoal is degradable and can be transformed from macro to micro particles under continued disturbance (e.g. soil management)³⁷. This explains the accumulation of microcharcoal at intermediate depths and implies that the earliest input events identified here are, in fact, significantly older”.

The fact that the ADE profiles contain more organic matter derived from C4 plants is significant to the story but unfortunately the ¹³C values in the ADE and Ultisol are too similar to be interpretable as a mixing of C3 and C4 plants – it could easily be due to differences in water use efficiency driven by the different water holding capacities of the soil. But if you believe that you can pick apart a plant signature between del ¹³C of -24 and -26: Was that C4 contribution derived from savanna grass or from corn grown in the ADE area? The interpretation put forth here relies on work showing that elsewhere in Amazonia there are/were patches of savanna that contributed C4 signature to soils. That discussion requires more explication than is provided here – really needs its own paragraph to lay out the argument because it seems as though the simplest explanation is the corn crop contribution – so why make it more complicated? Is it really legitimate to point to savanna sites farther upstream where it is more likely that most of the soils are more nutrient rich (at least according to the papers by Quesada et al)? I know these savanna patches and their potential significance to past conditions have been deemed important to the paleoecological relationships in the basin but how clear is the relationship? The relationship between upstream patches and the ADE site under consideration here is key to the argument but not well enough explicated.

To calculate the contribution of trees and grasses for the total carbon pool at different points in time the ¹⁴C age-dept model used in combination with typical biomass $\delta^{13}\text{C}$ signatures of trees ($-25.9\text{‰} \pm 0.4\text{‰}$) and C₄ grasses ($-11.2\text{‰} \pm 0.2\text{‰}$), which dominate forests and savannas, respectively. This approach,

which is standard previous in studies of this kind, is explained in the methods sections supported by several recent references of studies conducted in central Brazil (e.g. ⁷⁶⁻⁷⁸). To be consistent with previous studies, the proportion of C₄-derived is expressed as the relative (percentage) of grass input to the total SOC pool, which is statistically different between ADE and Ultisols in deep layers, but not in shallow layers. We argue that something happened and that something is in all likelihood a change in C₄ carbon input. As explained in the methods, other sources of variation, such as changes in water-use efficiency in response to elevated atmospheric CO₂, are not relevant at the temporal scale studied here, as this effect would be much smaller than differences between the C₃ and C₄ end members. Furthermore, we accounted for the diagenetic effects that enrich δ¹³C values after plant deposition. Finally, our interpretation of SOC δ¹³C ratios is consistent with those of many other geomorphological reconstructions near our site. We improved our explanation and added new references to show that *“the timing and magnitude of deposition at our site more closely match those found in sedimentary deposits that can be traced to early-Holocene open vegetation fires upstream, on the basis of ¹⁴C dates and δ¹³C signatures⁴⁴. Other paleoflood archives (i.e. soil biogenic silica)⁴⁵ and records of monsoon intensity (i.e. speleothem oxygen and strontium isotopes)⁴², indicate a climate-driven shift in river dynamics following a regional increase in precipitation after a persistent dry period (~8000 to ~4000 years ago)⁴⁶...”*

The material from L. 165-177 is presented as an explication of the savanna story and related ideas about depositional environments but it is too compact to provide the context necessary to understand the problem. For instance, I do not understand the quote about sediment deposition being activated or deactivated by specific environments. What landscape positions are being talked about? Is this in the floodplain (Varga forest) or in the upland (Terra Firmae forest)? And since there has not been an explicit discussion about the landscape position of the study location I can't envision the process being called on. From other reading (Denevan, Glaser, etc) I assume that we are talking about bluff sites that are in close proximity to the floodplain but I don't know that to be the case. Is the paleochannel being discussed here envisioned as contemporaneous with the process dynamics going on in the floodplain or is it etched into the upland? And what is the relevant time frame? – We know that the soils have different clay contents, is that because of different ages between the “levee” and the Ultisol site? An understanding of the stratigraphy at the contact between the two different landforms would be helpful here. I guess maybe the problem here is that the results are mixed with the discussion and the latter is not well enough developed.

We improved this section to clarify our point. As a continuation of the paragraph above, we now explain that *“savannas and grasslands of central Amazonia are associated with sediment deposits in dynamic landscapes representative of flood regimes that were either deactivated during the Holocene or are presently in the process of deactivation (i.e. after deactivation and exposure sedimentary deposits become suitable habitats for open ecosystems within the forest)⁴⁸. Fire frequency and charcoal production in those open ecosystems are orders of magnitude higher than in non-flammable rainforests⁴⁹⁻⁵¹, and thus C₄-derived charcoal accumulates in situ and in lowlands downstream”*.

L. 209-211. In Fig 6; left panel, what do the notations inside the graphs mean? And the Sr isotope Y-axis is screwed up. Somehow the graphic on the right side seems to have a lot of extraneous information that is beside the point – but maybe that is just because the depositional model the ms wants to get across is not well enough presented to include that material? Within reason I find the Sr and Nd isotope data to be convincing that there is a riverine input and that perhaps it is better to call on direct input of sediment than on the importation of fish bones although the Sr/Ca ratios could be jacked by thousands of years of differential leaching. Although it looks like there might not be much statistical power for Nd. Still I am having a hard time imagining this fluvial input being superimposed on the Ultisol soil/landscape rather than having two different sedimentary deposits separated by enough time to allow the Ultisol to have evolved to its present state of clay concentration whereas the “levee” deposit has evolved from a relatively sandy deposit to a sandy clay one. A focus on clay content rather than clay mineralogy. What is the particle size of the sediment that has supposedly been deposited on the levee (that derived from the

suspended sediment load in the river? How does this augmentation actually work? What are the relative topographic positions of the Ultisol and ADE? What does the mineralogy of the sand or silt fraction tell you about sedimentation? Too many questions here to allow an unambiguous interpretation of the isotope data. Also not to be too argumentative or fanciful here but what about Denevan's bluff model of habitation assertion that the location of habitation had to balance several features of the landscape: need to be close to the river and need to be close to fresher sediment that could be carried to the ADE sites for mixing with the existing soils to enhance nutrient status. The latter is perhaps more of an Anthropological construct than reality, but I do think that the isotope data and particle-size data could be simpatico with that model as well. And that plays into the first sentence in the following Human Inputs section. Are you implicitly discounting the model that humans added fresher sediment to the ADEs? If the argument here is that natural conditions drive the difference between ADE and Ultisol then that model probably needs to be addressed head on.

To answer these questions, we added a new paragraph in the methods and extracted the closest ICESat-2 points to the study site and compared the best estimate of terrain height to the SRTM elevation data in Google Earth (new Fig 7). The bluff heights are at a maximum of 27 m above the river level, which is above the 100-year flood now, but well within mid Holocene flooding regime (Passos et al). We added a new paragraph to the methods explaining that *"The site is located just north of the Solimões River near its confluence with the Negro River, within the sedimentary basin of the Amazon River. Satellite and field measurements show that the terrace is located ~40 m above the modern sea level, at a maximum of ~27 m above the modern river level, or ~10 m above the adjacent flood plain (3° 15' 6.8" S and 60° 13' 46.9" W)".* Additionally, we accepted this reviewer's suggestion regarding the bluff model, explaining that *"The relative topographic positions of ADE and Ultisol at the site matches the classic "bluff model" of prehistoric riverine settlement (i.e. close to the river and with fresh sediment mixing with the existing soil providing enhanced nutrient status)¹⁷. A bluff is indeed visible at the southwestern limits of the ADE patch, which is interpreted as a natural levee formed due to Holocene paleo river dynamics (Fig 1). This interpretation is based on other features of the Solimões-Amazon fluvial system, between the Negro and Madeira tributaries. Specifically, records of paleo river dynamics in many extensions of the Solimões River indicates that the combined action of neotectonics and rainforest establishment related to the increase of humidity in Amazonia ~6000 years ago allowed for sediment accumulation and asymmetric distribution of fluvial terraces throughout the region⁵⁹."*

L. 244-245. Is this a fair summary of the ideas (several centuries of ADE formation) in the literature about longevity and size of habitation? – seems like there is a lot of debate about this. For instance, Maezumi et al 2018 put the development of ADE technology at ~2500 years ago (at least in eastern Amazon). Perhaps a better way to discuss the results would be in terms of how they shed light on the existing ideas along with the idea that the ADEs are pre-existing features that were utilized by people.

We combined this suggestion with those of other reviewers to improve this section.

L. 264-266. The model proposed here for the "natural process" control and development of ADEs is not compelling. It likely due to too short presentation. Need to carefully lay out the argument. And it might be better to go to more of a results section and a discussion section format to drive that communication.

We combined this suggestion with those of other reviewers to improve this section.

REVIEWER COMMENTS

Reviewer #1 (Remarks to the Author):

Thank you to Silva et al. for the detailed consideration of the reviewer comments. The revised version of the manuscript by is much improved and engages more with the local and regional archaeological literature.

The authors now present a more holistic argument that the ADE soils were not the result of maize agriculture in the region. A recent study by Macedo et al. 2019: <http://dx.doi.org/10.1590/1981-81222019000100013> on soils from EMBRAPA came to similar conclusions:

"Consequently, the presence of typical ADEs in fertile floodplains is strong evidence that the soil was not intentionally altered for agriculture, since these regions are naturally fertile and contain nutrient levels far above those needed for to cultivate the most common plants (Havlin et al., 2003). These findings show that the formation of ADEs, at least initially, was not intentional for agricultural practices, disproving hypotheses related to the role of limiting natural factors in the establishment of permanent and sedentary settlements in pre-Columbian Amazonia." Macedo et al. 2019.

Macedo et al. and the majority of recent archaeological literature (summarized nicely by Schmidt et al. 2014 (<http://dx.doi.org/10.1016/j.jas.2013.11.002>), attribute the pre-agriculture ADE sites to deposits from residential contexts and in many cases, are products of waste disposal activities versus intentional agronomical uses to improve soil for cultivation. The connection with agriculture developed later on, as these soils were already formed. Macedo et al. (2018-2019) and others, argue ADEs were not intentionally formed for cultivation purposes. This interpretation is supported by the presence of faeces, plant and fish waste coupled with the absence of crop phytoliths in the EMBRAPA soils.

Regarding the Masters theses on the archaeology from the EMBRAPA region: please see the recent peer reviewed summary by Mongeló et al. 2020. <https://doi.org/10.1590/2178-2547-bgoeldi-2019-0079>.

Mongeló et al. document Early-Mid Holocene occupation in the region by hunter-fisher-gather communities. In my opinion, the more parsimonious explanation of the EMBRAPA ADE soils is from a domestic context and the result of human and food waste from early occupants subsisting largely on aquatic resources. See Teotonio (Watling et al. 2018, 2019) as an example of early fisher-forager communities. The needed contribution of fish to enrich the ADE soils is already discussed in detail by Silva et al. If the interpretation that the EMBRAPA ADE is of human origin is considered, the EMBRAPA ADE site represents some of the earliest evidence of permanent fisher-hunter-gatherer communities in the Amazon and may have important implications for understanding the transition from nomadic to sedentary populations.

The Later min- to late- Holocene ADEs in other sites that demonstrate the presence of crop phytoliths, were later exploited by people for agriculture concurrent with the timeline discussed by Bush et al. and included in the discussion by Silva et al. This interpretation is supported by increasing evidence of crop cultivation prior to the development of ADEs (many of these papers are now included by Silva et al.), suggesting ADEs were not a requisite for mid-Holocene crop cultivation but were often exploited to increased subsistence yields with the Late Holocene expansion of crop cultivation in the Amazon.

*NB: There may be exceptions to the intentional formation of ADEs for crop cultivation at unique sites including Santarém (Stenborg et al. 2016) and Teotonio (Watling et al. 2018, 2019), but the majority of the recent ADE archaeological literature suggest this is not the norm.

Regarding Late Glacial-Early Holocene flood levels and contribution of fluvial material into the ADE soils (13 to 10 k; Passos et al.) (Lines 226-228). As Silva et al. argue, most of the documented ADE

sites are located on high terraces 30 to 40 m above the floodplains. However, not all of these bluff sites would have been flooded during the Holocene. Additionally, there is also an exponential increase in the number of ADE sites in the last 2k on the bluffs but modern flood levels stabilize around 6k. This holds true even with rivers such as the Madeira that experience increased river volume during the Late Holocene.

If the authors argue the EMBRAPA ADE is not attributed to residential contexts from Early Holocene hunter-fisher-gather occupation in the area, please clarify the rationale as to why the EMPRAPA bluff site would have a different developmental trajectory than later ADE bluff sites that develop across the Amazon in the last 2k.

Minor comment:

Line 187: Typo: change me to be.

Reviewer #2 (Remarks to the Author):

I have read the response to all the issues raised by the reviewers and it is complete and thoughtful. I have also read the revised manuscript, and believe that the authors have adequately addressed the issues raised in my review and also in the other reviews. As a result I find this version of the manuscript acceptable for publication. I have appended pdf version with comments, but these are all of a minor grammatical/typographical nature.

Something that hadn't occurred to me until I re-read the manuscript is that it will be of major significance to the massive effort that has gone in the 'biochar' over the last decade. It's always been clear that ADEs were 'more than just charcoal' but I think this underscores the fact and greatly extends understanding. It doesn't render biochar research meaningless, but it does possibly suggest that biochar research might need to incorporate a broader view to optimise use in modern agriculture. This, along with the archaeological interest does make the manuscript of broad significance.

Reviewer #3 (Remarks to the Author):

I found the revised ms to answer my concerns which primarily had to do with lack of clarity around some of the conceptual models that were presented. Also the figures have been greatly improved. I have no further concerns with the presentation of the ms and think that it is an important data-driven contribution to the ADE literature.

address specific reviewer requests or find any points invalid, please explain why in the point-by-point response.

Below you will find a point-by-point response to all major comments and suggestions. We have also enclosed an annotated version of the revised manuscript, which highlights all areas of improvement including new calculations, references, and data interpretation.

Response to Reviewer 1

Thank you to Silva et al. for the detailed consideration of the reviewer comments. The revised version of the manuscript by is much improved and engages more with the local and regional archaeological literature.

Thank you for the positive evaluation of our revised submission.

The authors now present a more holistic argument that the ADE soils were not the result of maize agriculture in the region. A recent study by Macedo et al. 2019: <http://dx.doi.org/10.1590/1981-81222019000100013> on soils from EMBRAPA came to similar conclusions: “Consequently, the presence of typical ADEs in fertile floodplains is strong evidence that the soil was not intentionally altered for agriculture, since these regions are naturally fertile and contain nutrient levels far above those needed for to cultivate the most common plants (Havlin et al., 2003). These findings show that the formation of ADEs, at least initially, was not intentional for agricultural practices, disproving hypotheses related to the role of limiting natural factors in the establishment of permanent and sedentary settlements in pre-Columbian Amazonia.” Macedo et al. 2019.

Thank you for bringing this reference to our attention. We added this reference to our discussion, which helped build a stronger case for our core hypothesis and data interpretation.

Macedo et al. and the majority of recent archaeological literature (summarized nicely by Schmidt et al. 2014 (<http://dx.doi.org/10.1016/j.jas.2013.11.002>), attribute the pre-agriculture ADE sites to deposits from residential contexts and in many cases, are products of waste disposal activities versus intentional agronomical uses to improve soil for cultivation. The connection with agriculture developed later on, as these soils were already formed. Macedo et al. (2018-2019) and others, argue ADEs were not intentionally formed for cultivation purposes. This interpretation is supported by the presence of faeces, plant and fish waste coupled with the absence of crop phytoliths in the EMBRAPA soils.

We agree with this assessment and we added a new paragraph to the ‘Human Inputs’ section, where we discuss the “non-intentional” genesis hypothesis proposed by Schmidt and Macedo. Here is a new passage that cites both references:

*“In pre-Columbian times, the most likely human-derived nutrient inputs to soil were food waste and faeces. Archaeologists hypothesize that such domestic waste deposits led to the widespread formation of anthropic soils near ancient settlements²⁰, in a process that some authors have described as “non-intentional”⁵⁸. Molecular markers of human faeces found in several ADEs, including our site²¹, offer support for that hypothesis. However, we still don’t know how much waste would have been needed to form a typical ADE patch. Here, we offer a first approximation based on the excess amount of P and Ca, calculated in relation to the surrounding Ultisol at our site (~32 Mg ha⁻¹; **Fig 3**), which implicitly represents the depositional history of many other elements that correlate strongly with those macronutrients (**Fig S1**).”*

Regarding the Masters theses on the archaeology from the EMBRAPA region: please see the recent peer

reviewed summary by Mongeló et al. 2020. <https://doi.org/10.1590/2178-2547-bgoeldi-2019-0079>. Mongeló et al. document Early-Mid Holocene occupation in the region by hunter-fisher-gatherer communities. In my opinion, the more parsimonious explanation of the EMBRAPA ADE soils is from a domestic context and the result of human and food waste from early occupants subsisting largely on aquatic resources. See Teotonio (Watling et al. 2018, 2019) as an example of early fisher-forager communities. The needed contribution of fish to enrich the ADE soils is already discussed in detail by Silva et al. If the interpretation that the EMBRAPA ADE is of human origin is considered, the EMBRAPA ADE site represents some of the earliest evidence of permanent fisher-hunter-gatherer communities in the Amazon and may have important implications for understanding the transition from nomadic to sedentary populations.

Thank you for bringing this reference to our attention. We agree that this study is relevant and, although it was conducted in western Amazonia thousands of kilometers away from our site, we added this reference as a possible alternative explanation to our interpretation. Here is one passage where we cite this new study, adding some of the language suggested above:

“... It is possible that dense populations occupied riverine areas uninterrupted for thousands of years longer than previously recorded, contributing large amounts of fish waste to the soil, as it appears to have been the case for some of the first hunter-fisher-gatherer communities in western Amazonia⁶⁵. It is also possible that those communities were capable of recycling nearly all of their waste into ADEs, with little to no loss through leaching or burning. It is more plausible, however, that human-derived inputs represent a minor fraction of ADE’s chemical makeup, a fraction that, we hypothesize, was introduced in the relatively recent past. If our hypothesis holds true, other sites will exhibit similar depositional patterns and biogeochemical signatures as those reported here ... Other ADEs should be investigated for a potential alluvial origin on the basis of physical properties and elemental sources, which may have important implications for understanding the transition from nomadic to sedentary populations and their effect on regional ecosystems”

The Later min- to late- Holocene ADEs in other sites that demonstrate the presence of crop phytoliths, were later exploited by people for agriculture concurrent with the timeline discussed by Bush et al. and included in the discussion by Silva et al. This interpretation is supported by increasing evidence of crop cultivation prior to the development of ADEs (many of these papers are now included by Silva et al.), suggesting ADEs were not a requisite for mid-Holocene crop cultivation but were often exploited to increase subsistence yields with the Late Holocene expansion of crop cultivation in the Amazon.

*NB: There may be exceptions to the intentional formation of ADEs for crop cultivation at unique sites including Santarém (Stenborg et al. 2016) and Teotonio (Watling et al. 2018, 2019), but the majority of the recent ADE archaeological literature suggest this is not the norm.

We agree. ADEs were not a requisite for mid-Holocene crop cultivation but were often exploited to increase subsistence yields with the Late Holocene expansion of crop cultivation in the Amazon. Our revised manuscript discusses other forms of occupation and agriculture to make that point abundantly clear.

Regarding Late Glacial-Early Holocene flood levels and contribution of fluvial material into the ADE soils (13 to 10 k; Passos et al.) (Lines 226-228). As Silva et al. argue, most of the documented ADE sites are located on high terraces 30 to 40 m above the floodplains. However, not all of these bluff sites would have been flooded during the Holocene. Additionally, there is also an exponential increase in the number of ADE sites in the last 2k on the bluffs but modern flood levels stabilize around 6k. This holds true even with rivers such as the Madeira that experience increased river volume during the Late Holocene.

We agree. We have now improved our discussion to explain that proximity to the river and elevation are strong predictors of ADEs, but do not tell full story. For example we added the following sentences:

“... This could explain why maximum entropy algorithms show a “highly predictable” regional pattern of ADE distribution based solely on climatic and geomorphological parameters, such as distance from rivers and elevation². Our findings offer a window into the past and underscore the need for a broader view of landscape evolution as a path towards understanding Anthrosol formation and improving applications, such as those arising from biochar research.”

If the authors argue the EMBRAPA ADE is not attributed to residential contexts from Early Holocene hunter-fisher-gather occupation in the area, please clarify the rationale as to why the EMPRAPA bluff site would have a different developmental trajectory than later ADE bluff sites that develop across the Amazon in the last 2k.

We chose to not speculate about the origin of other sites; however, we now explain that our data provide a roadmap for other ADEs to be investigated for an alluvial origin. Specifically, we added several sentences that offer a more explicit explanation of our title choice *“A new hypothesis for the origin of Amazonian Dark Earths”* followed by pointed suggestions that will serve as targets for future investigations. For example, we expect that the timing of formation of other ADEs would be pushed back significantly following a detailed analysis of microcharcoal deposition. To the best of our knowledge, our study is the first to provide detailed analysis of nutrient and microcharcoal source and time of deposition in the region. Therefore, we offer specific suggestions that we hope will stimulate future research in the region. Here are some examples of new sentences added for that purpose:

“We hypothesize that indigenous peoples exploited the exceptional fertility of ADEs but were not directly involved in their genesis, a hypothesis that remains to be tested at other sites.”

“We posit that the timing of formation of other ADEs would be pushed back significantly following a detailed analysis of microcharcoal deposition.”

“If our hypothesis holds true, other sites will exhibit similar depositional patterns and biogeochemical signatures as those reported here.”

“Other ADEs should be investigated for a potential alluvial origin on the basis of physical properties and elemental sources, which may have important implications for understanding the transition from nomadic to sedentary populations and their effect on regional ecosystems.”

“Future studies stand to gain valuable information from interdisciplinary approaches that combine indigenous knowledge of landscape development with biogeochemical and geographical parameters to inform prioritization for sustainable land use in the Amazon and other tropical environments”.

Minor comment:

Line 187: Typo: change me to be.

Corrected

Reviewer #2 (Remarks to the Author):

I have read the response to all the issues raised by the reviewers and it is complete and thoughtful. I have also read the revised manuscript, and believe that the authors have adequately addressed the issues raised in my review and also in the other reviews. As a result I find this version of the manuscript acceptable for

publication. I have appended pdf version with comments, but these are all of a minor grammatical/typographical nature.

Thank you for the positive evaluation of our work. We have accepted all suggestions and made all changes included in the annotated pdf provided by the reviewer.

Something that hadn't occurred to me until I re-read the manuscript is that it will be of major significance to the massive effort that has gone in the 'biochar' over the last decade. Its always been clear that ADEs were 'more than just charcoal' but I think this underscores the fact and greatly extends understanding. It doesn't render biochar research meaningless, but it does possibly suggest that biochar research might need to incorporate a broader view to optimise use in modern agriculture. This, along with the archaeological interest does make the manuscript of broad significance.

We agree and added language to reflect that important implication of our findings. For example, we now state that:

“Our findings offer a window into the past and underscore the need for a broader view of landscape evolution as a path towards understanding Anthrosol formation and improving applications, such as those arising from biochar research.”

Reviewer #3 (Remarks to the Author):

I found the revised ms to answer my concerns which primarily had to do with lack of clarity around some of the conceptual models that were presented. Also the figures have been greatly improved. I have no further concerns with the presentation of the ms and think that it is an important data-driven contribution to the ADE literature.

Thank you for the positive assessment of our work.

REVIEWERS' COMMENTS

Reviewer #1 (Remarks to the Author):

I have read the response to the issues raised in the previous review which has addressed the majority of my concerns. Thank you to the authors for addressing the issues raised in my previous comments. Minor comments for the revised manuscript:

1. In addition to the pie charts for the pyrogenic microcarbon % values, can a line plot of the be added to the SI material of the microcarbon percentage with percentage on the x-axis and depth on y-axis and labels approximate ages next to the sample depths. Also can the A, B, C group lables be added to the age depth profile to facilitate the comparison of both depth and timing of changes across the different figures.

In Lines 202-203 can a sentence or so be added on pyrogenic microcharcoal interpretation in this record. Are the EMBRAPA microcharcoal data though to be produced in situ or washed in during flood events and are these grass fires thought to be natural (e.g. lightning caused or anthropogenic in orgin?)

Minor comments:

Define microcharcoal size range at first use (e.g. line 89)

Line 86: replace best with a well studied...

Line 163: Define macro and microcharcoal size classes

Line 288: Rephrase this sentence, this is with in the range of prefvioulsy population density etimates as stated in Line 294.

Line 260: replace don't with do not

Line 316: replace best with well studies.

REVIEWERS' COMMENTS

I have read the response to the issues raised in the previous review which has addressed the majority of my concerns. Thank you to the authors for addressing the issues raised in my previous comments.

Minor comments for the revised manuscript:

1. In addition to the pie charts for the pyrogenic microcarbon % values, can a line plot of the be added to the SI material of the microcarbon percentage with percentage on the x-axis and depth on y-axis and labels approximate ages next to the sample depths. Also can the A, B, C group labels be added to the age depth profile to facilitate the comparison of both depth and timing of changes across the different figures.

We accepted the first part of this suggestion and added two new graphs to the supplementary materials (SI Figure 2) showing microcharcoal and $\delta^{13}\text{C}$ trends with depth. The original data for all figures is available in the permanent repository listed at the end of the document. We did not include the suggested A, B, C to the age-depth plot because those depth ranges were not used in the age-depth model; that is, unlike in the pie chart insets where we showed average microcharcoal amounts, we report the individual measurements in the age-depth figure and therefore the indication of depth ranges would be confusing in this case.

In Lines 202-203 can a sentence or so be added on pyrogenic microcharcoal interpretation in this record. Are the EMBRAPA microcharcoal data thought to be produced in situ or washed in during flood events and are these grass fires thought to be natural (e.g. lightning caused or anthropogenic in origin?)

Added as suggested

Minor comments:

Define microcharcoal size range at first use (e.g. line 89)

Corrected as suggested

Line 86: replace best with a well studied...

Corrected as suggested

Line 163: Define macro and microcharcoal size classes

Corrected as suggested

Line 288: Rephrase this sentence, this is within the range of previously population density estimates as stated in Line 294.

Corrected as suggested

Line 260: replace don't with do not

Corrected as suggested

Line 316: replace best with well studies.

Corrected as suggested